# Host-diet-gut microbiome interactions influence human energy balance: a randomized clinical trial

Karen D. Corbin [1], Elvis A. Carnero [1], Blake Dirks [2,3], Daria Igudesman [1], Fanchao Yi[1], Andrew Marcus[2,4], Taylor L. Davis[2,3], Richard E. Pratley[1], Bruce E. Rittmann [3,5], Rosa Krajmalnik-Brown [2,5] ✉ & Steven R. Smith[1] ✉

The gut microbiome is emerging as a key modulator of human energy balance. Prior studies in humans lacked the environmental and dietary controls and precision required to quantitatively evaluate the contributions of the gut microbiome. Using a Microbiome Enhancer Diet (MBD) designed to deliver more dietary substrates to the colon and therefore modulate the gut microbiome, we quantified microbial and host contributions to human energy balance in a controlled feeding study with a randomized crossover design in young, healthy, weight stable males and females (NCT02939703). In a metabolic ward where the environment was strictly controlled, we measured energy intake, energy expenditure, and energy output (fecal and urinary). The primary endpoint was the within-participant difference in host metabolizable energy between experimental conditions [Control, Western Diet (WD) vs. MBD]. The secondary endpoints were enteroendocrine hormones, hunger/satiety, and food intake. Here we show that, compared to the WD, the MBD leads to an additional 116 ± 56 kcals (P < 0.0001) lost in feces daily and thus, lower metabolizable energy for the host (89.5 ± 0.73%; range 84.2-96.1% on the MBD vs. 95.4 ± 0.21%; range 94.1-97.0% on the WD; P < 0.0001) without changes in energy expenditure, hunger/satiety or food intake (P > 0.05). Microbial 16S rRNA gene copy number (a surrogate of biomass) increases (P < 0.0001), beta-diversity changes (whole genome shotgun sequencing; P = 0.02), and fermentation products increase (P < 0.01) on an MBD as compared to a WD along with significant changes in the host enteroendocrine system (P < 0.0001). The substantial interindividual variability in metabolizable energy on the MBD is explained in part by fecal SCFAs and biomass. Our results reveal the complex host-diet-microbiome interplay that modulates energy balance.

Microbial communities in the colon have a profound effect on host physiology, including immune function, inter-organ communication, and metabolism[1]. The majority of studies in humans have correlated the gut microbiota's composition, gene expression, and metabolism with human health endpoints such as body weight, glycemic control, and inflammatory bowel diseases[2,3]. What remains to be determined is whether the gut microbiome is a causal driver or merely a reflection of host physiology[4].

The effect of the gut microbiome on weight regulation has been a topic of high interest[5]. Obesity is a major public health concern that is at the nexus of metabolic diseases such as cardiovascular disease, non-alcoholic fatty liver disease, and type 2 diabetes[6]. The gut microbiome has emerged as a control center for host energy balance through its impacts on energy harvest from food, gut hormones, and signaling through metabolites such as short chain fatty acids (SCFAs)[5]. Existing data are largely restricted to preclinical models or observational studies[7–9]. Prior controlled feeding studies have demonstrated that high-fiber diets are associated with reduced host metabolizable energy[10] and that varying dietary energy load can alter energy harvest efficiency in a way that correlates to phyla in the gut microbiota[11]. Despite these advances, studies to date lack a comprehensive quantitative evaluation of the contribution of the gut microbiome to the entire energy balance equation, including energy intake, energy expenditure, and fecal energy losses. Prior studies were also insufficiently precise to detect potentially modest differences in the composition of the gut microbiome, which can vary dramatically between individuals, particularly when appropriate environmental controls were not implemented.

To address these critical knowledge gaps, here we describe the results of the intersection of host-diet-gut microbiome factors on human energy balance generated by performing a controlled feeding study in a metabolic ward using a deep-phenotyping paradigm of quantitative bioenergetics (NCT02939703)[12] (Supplementary Fig. 1). The primary endpoint for the protocol was the within-participant difference in 24-h fecal energy normalized to the total daily energy intake measured during the 6-day calorimetry block within each domiciled diet period. We hypothesized that with the MBD, there would be higher fecal energy (and thus lower host metabolizable energy) due to the greater availability of dietary substrates to the colonic gut microbes[13]. This hypothesis also was supported by our in silico mathematical model[14] that predicted an additional 110 kcals of additional fecal energy loss on the MBD per 2000 kcal consumed as compared to the WD. The principal secondary endpoints tested hypotheses about how diet-induced changes in the gut microbiota might change enteroendocrine hormone secretion, hunger/satiety, and food intake. These measures were evaluated during the final two days of each domiciled diet period. With this paradigm, we find that delivery of more dietary substrates to the gut microbiome leads to a net negative energy balance that is accompanied by a robust remodeling of gut microbiota composition, diversity and function and changes in host enteroendocrine hormones.

## Results

### Validation of study paradigm
The details of participant flow from enrollment through analysis are detailed in Supplementary Fig. 2. The intervention implemented in this trial included a highly digestible control Western Diet (WD) and a Microbiome Enhancer Diet (MBD). The MBD maximized the availability of dietary substrates to the gut microbiome and included four dietary drivers: dietary fiber, resistant starch, large food particle size, and limited processed foods (Supplementary Fig. 1). Our design provided equivalent dietary metabolizable energy (kcal) and total macronutrients (fat, protein, carbohydrates) based on classic principles and equations of food digestibility[15]. Diets were prepared in our metabolic kitchen and validated by measuring energy content via chemical analysis. Analysis of intake during the nine domiciled days that provided meals exactly as designed (i.e., excluding an ad libitum feeding day and a gastric emptying test day which required a liquid meal) demonstrated that the diets consumed by study participants delivered the planned energy, macronutrients, and gut microbiome dietary drivers (Supplementary Table 1).

To avoid the confounding effects of over- or underfeeding on host and microbial metabolism, we evaluated energy balance by real-time assessment of energy intake (personalized to the energy needs of each participant[12]) and energy expenditure (measured via whole-room indirect calorimetry). We found that energy balance was maintained within our target of ±50 kcals per 6-day calorimeter stay (WD 4.1 ± 5.1 kcal/day; MBD 5.4 ± 2.8 kcal/day; $P = 0.8$; Supplementary Fig. 3a). Weight stability was a secondary criterion for evaluating energy balance, and we previously reported that weight was stable during the 6-day calorimetry assessment period whilst the primary endpoint was measured and data were being generated without a link to diet assignment[12].

Surveillance of adverse events revealed minimal gastrointestinal or other side effects that did not differ by diet (Supplementary Data 1). Adherence was equivalent between diets during the metabolic ward period (99.6 ± 0.19% on MBD vs. 99.9 ± 0.10% on WD, $P = 0.27$; Supplementary Fig. 3b). Therefore, our validated paradigm was well tolerated by study participants.

### Participant characteristics
Young, healthy, weight-stable individuals were enrolled to quantify whole-body bioenergetics without the confounding effects of age and metabolic disease[16] and to establish the comparative data needed for future studies enrolling people with various health conditions (Table 1). We excluded people with recent antibiotic use or chronic health conditions that were evaluated by medical history, physical exam and standard clinical labs. The study sample was 30.8 ± 1.9 years of age, with a BMI within the normal weight to overweight range (Table 1). All participants reported normal stool patterns based on the Bristol Stool Scale[17] and sleep duration of 5.95 ± 0.32 h (Table 1). The habitual self-reported free-living intake of total dietary fiber was

**Table 1 | Baseline characteristics**

| Total N | 17 |
|---|---|
| Age (years) | 30.8 ± 1.9 |
| BMI (kg/m²) | 25.1 ± 0.52 |
| Female Sex | 8 (47.1) |
| Race | |
| Black | 11 (64.7) |
| White | 5 (29.4) |
| Unknown | 1 (5.9) |
| Hispanic/Latino Ethnicity | 6 (35.3) |
| Weight (kg) | 70.5 ± 3.0 |
| Waist to Hip Ratio | 0.83 ± 0.02 |
| Bristol Stool Scale[a] | 3.8 ± 0.10 |
| Type 3 | 3 (17.65) |
| Type 4 | 14 (82.35) |
| HbA1c (%) | 5.0 ± 0.09 |
| TSH (u[IU]/mL) | 1.7 ± 0.19 |
| AST (units/L) | 25.3 ± 2.4 |
| ALT (units/L) | 22.3 ± 3.4 |
| BUN (mg/dL) | 11.7 ± 0.64 |
| Creatinine (mg/dL) | 0.98 ± 0.06 |
| Sleep (hours) | 5.95 ± 0.32 |
| Free-living dietary intake | |
| Carbohydrates (%) | 51 ± 3 |
| Fat (%) | 34 ± 2 |
| Protein (%) | 16 ± 2 |
| Fiber (g/1000 kcal) | 7.6 (6.6, 10.4) |

Continuous variables reported as mean ± s.e.m or median (IQR). Categorical variables reported as N (%).

[a]The Bristol Stool Scale evaluates stool type based on shape and consistency, with scores of 3-4 indicating neither constipation nor loose stools[17].

generally low in our study sample (7.6 g/1000 kcal/day [IQR 6.6, 10.4 g/1000 kcal/day]) on par with a Western diet and representative of adults living in the United States[18]. Free-living intake of macronutrients was similar to those in the study diets (Table 1). The total sleep period during the 6-days in the calorimeter (when our primary endpoint was measured) was held constant between diets (8 h; Supplementary Table 2), which is important because sleep duration impacts hunger, circadian rhythms and downstream host and microbial phenotypes[19]. With our radar-based motion detector, we calculated motion-free sleep, which is a surrogate of high-quality sleep that we use to minimize the effects of small amounts of involuntary motion during sleep on sleep energy expenditure[20]. We found that sleep was not different between the two diet conditions during the calorimetry stay (mean ± s.e.m. motion-free sleep duration was 3.5 ± 0.75 h on WD and 3.5 ± 0.5 h on MBD; Supplementary Fig. 3c)

### Diet modulated host metabolizable energy

The overall goal of our study was to modulate the gut microbiome and employ a quantitative paradigm with enough precision to detect within-participant responses to the diet intervention. A key contribution to understanding the role of the gut microbiome on energy balance involves fecal energy. Prior methods lack precision and often provide results as energy per gram of feces, which does not contextualize fecal energy in terms of dietary intake (which we precisely controlled) and makes it difficult to interpret the relationships of fecal energy to host phenotypes[21]. To this end, according to the method of

Pak[22], we administered a low, non-laxative dose of non-absorbable non-digestible polyethylene glycol (PEG) with each meal. We measured the PEG concentration in fecal samples to normalize each fecal measurement to 24-h based on expected daily PEG excretion. To quantify fecal energy loss, we used chemical oxygen demand (COD), a measure of electron equivalents in organic carbon[23] and adjusted the result to PEG recovery. COD is typically used for microbial bioenergetics in environmental biotechnology[16]. We previously reported that, for food items, COD correlates highly to the commonly used bomb calorimetry method ($R^2 = 0.97$)[24]. COD is a less expensive alternative that provides relevant information for microbial electron balances, and more physiologically relevant measurements since COD does not include the oxidation of ammonia, which humans do not utilize as an electron donor[23,24]. Additionally, COD is advantageous because it simultaneously measures electrons available to humans and microbes, thus enabling electron balances to quantify energy flow[23]. Based on this (fecal energy as COD adjusted to PEG recovery), the MBD increased mean daily fecal energy losses, compared to the WD, over the six calorimetry days of the domiciled controlled-feeding period (73.0 ± 6.1 gCOD/day on MBD vs. 32.1 ± 2.5 gCOD/day on WD; $P = 2.96 \times 10^{-7}$; Fig. 1a). When fecal energy loss was adjusted to total energy intake to calculate host metabolizable energy (primary endpoint), we found that it was lower with the MBD (89.5 ± 0.73% on the MBD vs. 95.4 ± 0.21% on the WD (Fig. 1b; $P = 2.73 \times 10^{-7}$), which equates to an additional 116 ± 56 kcals daily channeled to feces on the MBD vs. the WD (Fig. 1c; $P = 4.95 \times 10^{-7}$).

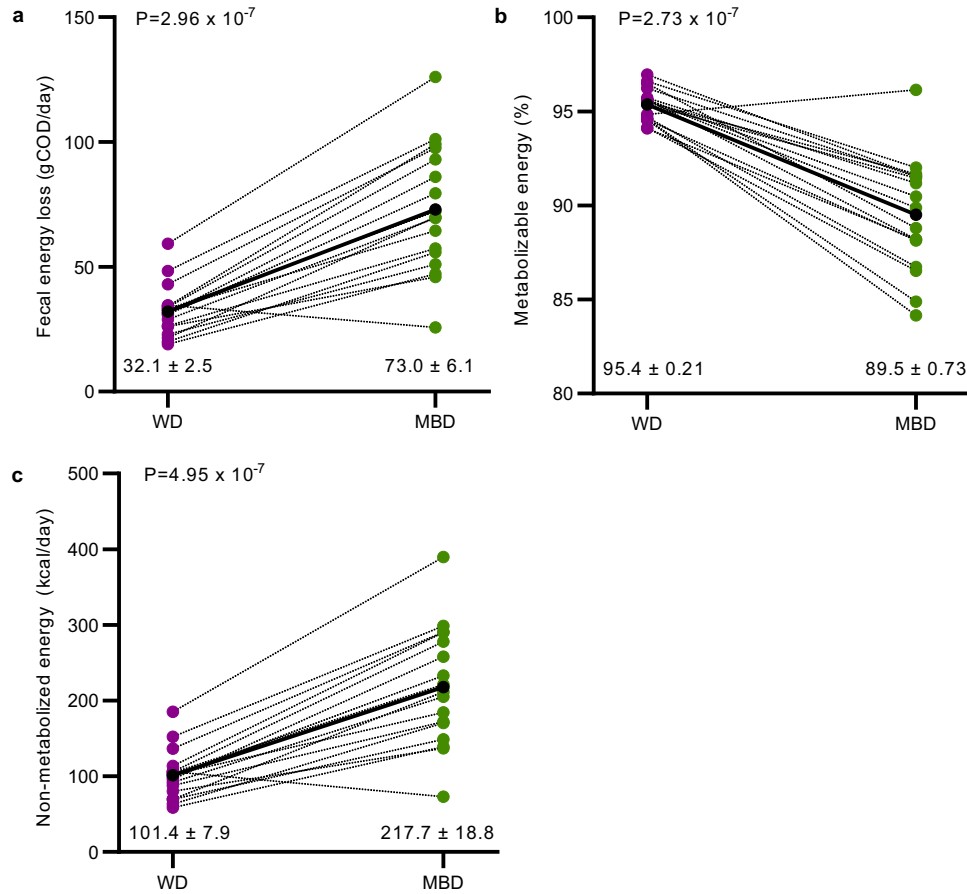

**Fig. 1 | The microbiome enhancer diet reduced host metabolizable energy.** **a** Daily energy lost by each participant in feces on the WD vs. MBD in grams COD/day (gCOD/day). **b** Host metabolizable energy based on the proportion of fecal COD to dietary intake. **c** Calculated host non-metabolizable energy (kcals). All data reported as are mean ± s.e.m. *n* = 17 per diet for all panels. *P* values are from linear mixed effects regression models and denote a statistically significant effect of diet on each endpoint. Source data are provided as a Source Data file. COD Chemical Oxygen Demand, MBD Microbiome Enhancer Diet (green), WD Western Diet (purple).

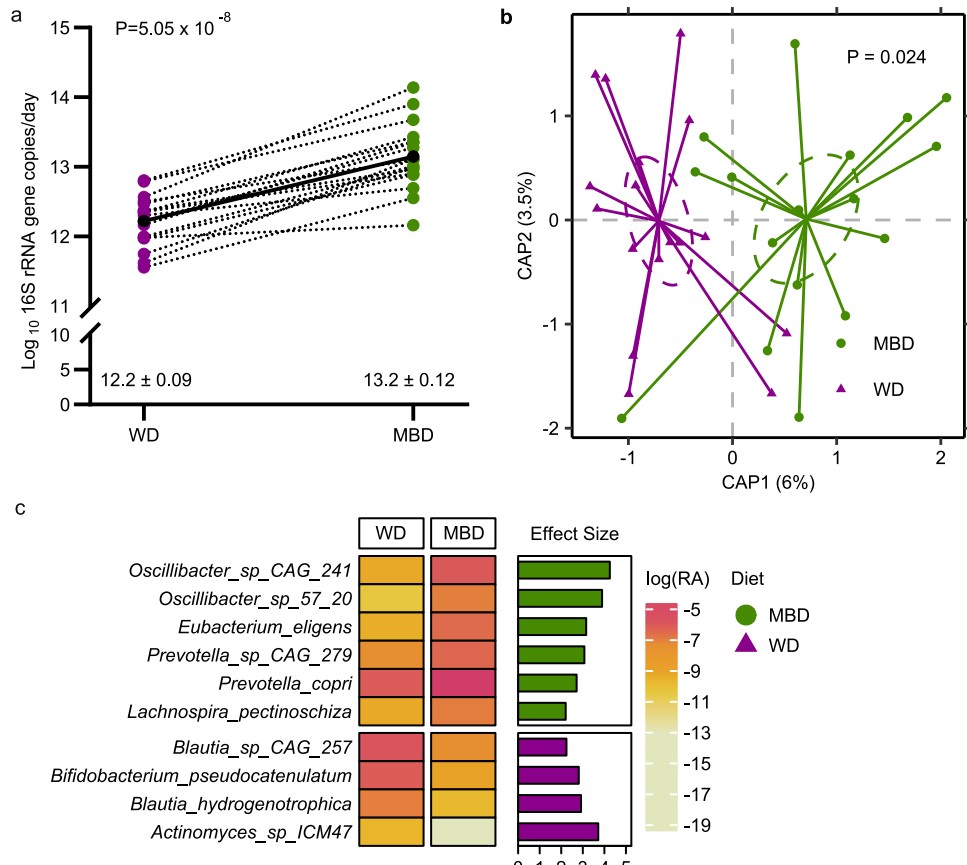

**Fig. 2 | Diet modulated the gut microbiome. a** Fecal bacterial 16S rRNA gene copy number (a surrogate of biomass); *P* value is from linear mixed effects regression model and denotes a statistically significant effect of diet on 16S rRNA gene copy number. **b** Beta-diversity (Bray−Curtis Dissimilarity). *P* value is from PERMANOVA test and denotes a statistically significant effect of diet on Bray−Curtis Dissimilarity metric. **c** Heatmap showing the natural-log-transformed mean relative abundance of species whose relative abundance was significantly different by diet (based on

MaAsLin2); bar plot shows the effect size of the regression coefficient from compound Poisson regression models comparing the relative abundance of each species by diet. Species shown in this figure were significantly different by diet (*P* values were corrected to produce *Q* values using the Benjamini−Hochberg method; *Q* < 0.05), and the diet difference had an effect size ≥2. *n* = 17 per diet for all panels. Source data are provided as a Source Data file. CAP Canonical Analysis of Principal Coordinates, MBD Microbiome Enhancer Diet (green), WD Western Diet (purple).

## Diet modulated the gut microbiome

Given our primary finding that the MBD produced a significant decrease in host metabolizable energy compared to the WD, thereby reducing energy available to the host, we next evaluated the microbial phenotype associated with host energy balance. Mean daily fecal weight was higher on the MBD ($P = 1.24 \times 10^{-5}$; Supplementary Fig. 4a), and a proportion of this additional weight was due to a significant increase in 16S rRNA gene copy number ($P = 5.05 \times 10^{-8}$; Fig. 2a), a surrogate of fecal bacterial biomass. Supporting this result, the MBD was predicted by our in silico mathematical model[25] to produce 19.6 ± 3.5 gCOD/day of microbial biomass compared to 9.4 ± 1.2 gCOD/day on the WD, which is >25% of the total energy content of feces on both diets.

Using whole-genome sequencing (WGS), we evaluated whether the increase in bacterial biomass was accompanied by a change in microbial diversity. Alpha-diversity assessed by species richness and evenness did not differ between diets (Supplementary Fig. 4b, c). In contrast, beta-diversity showed significant and stark separation by diet whether evaluated by Bray−Curtis (Dis)similarity ($P = 0.02$; Fig. 2b) or Jaccard Similarity ($P = 0.02$; Supplementary Fig. 4d).

To further explore the compositional changes in the microbiome associated with our two experimental diets, we derived regression coefficients testing differences in microbial species relative abundance (WGS) by diet using MaAsLin2's compound Poisson regression model, which adeptly handles zero-inflated data[26], and capitalizes on the

statistical power of the crossover design. Although relative abundance did not differ between the diets at the phylum and family levels (Supplementary Table 3), we found 53 differentially abundant taxa at the species level ($P < 0.05$; Supplementary Fig. 5a, b), of which 10 had a $Q < 0.05$ and differential effect size ≥2 (Fig. 2c). In accordance with dietary substrate availability, six species had a higher relative abundance on the MBD and included dietary fiber degraders (*Prevotella copri*, uncharacterized *Prevotella*, and *Lachnospira pectinoschiza*[27]; $Q = 1.46 \times 10^{-06}$, 0.0005, and 0.001, respectively) and/or butyrate producers [(*Lachnospira pectinoschiza*[28], *Eubacterium eligans*[29], and likely the uncharacterized *Oscillibacter* (CAG_241 and 57_20)[30] ($Q = 1.46 \times 10^{-06}$, 0.001, 7.44 ×$10^{-07}$, 0.01 and 2.27 ×$10^{-07}$, respectively)]. In contrast, the 4 species with a higher relative abundance on the WD included *Blautia hydrogenotrophica*, *Bifidobacterium pseudocatenulatum*, uncharacterized *Blautia CAG:257*, and uncharacterized *Actinomyces ICM7* ($Q = 0.006$, 5.6 ×$10^{-05}$, 0.001, 0.02, respectively). These four species derive their source of fermentation from host-glycans, simple sugars[31,32], or fermentation products generated by other gut microbes, mainly $CO_2$[33] and $H_2$[34]. As a means of validation, we repeated this analysis with ANCOM-BC[35–37], and retained the significance of most of the identified species in the signature (Supplementary Data 2).

The observed diet-induced changes in microbial composition were paralleled by an increase in fermentation, evidenced by higher SCFAs on the MBD vs. WD in feces (total, acetate, propionate, and butyrate; $P = 0.001$, 0.002, 0.007, and 0.0005, respectively; Fig. 3a)

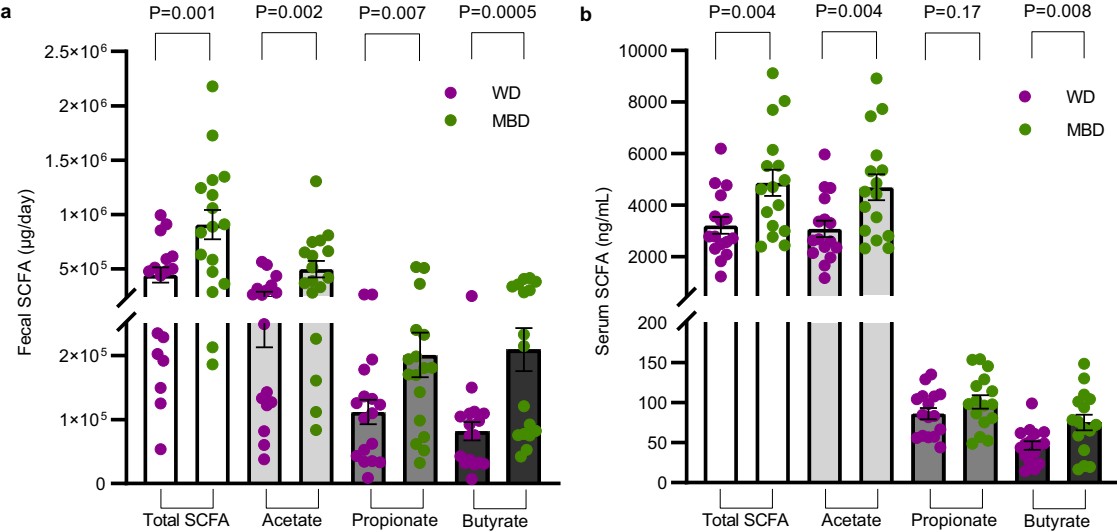

**Fig. 3 | Increased microbial fermentation on the microbiome enhancer diet.**
**a**, **b** Fecal and circulating short chain fatty acids. Data are presented as mean ± s.e.m ($n = 17$ per diet for panel **a** and $n = 16$ for panel **b**). Error bars in panels **a** and **b** are displayed as s.e.m. *P* values are from linear mixed effects regression models and denote a statistically significant effect of diet on fecal and serum SCFAs. Source data are provided as a Source Data file. MBD Microbiome Enhancer Diet (green), SCFA short-chain fatty acids, WD Western Diet (purple).

and serum (total, acetate, and butyrate; $P = 0.004$, 0.004, and 0.008, respectively; Fig. 3b). Thus, the microbiota signature that defined the response to the MBD i) channeled more energy to the microbes (instead of the host), ii) increased microbial fermentation, iii) increased fecal and serum SFCAs, and iv) increased biomass. In contrast, the WD led to conditions in which the gut microbes were "starved" because a higher proportion of metabolizable energy had been digested and absorbed by the host in the upper gastrointestinal tract.

## Host response to diet-gut microbiome interactions

We explored whether the differential host metabolizable energy was associated with changes in weight/body composition, gut motility, appetite, and/or hormonal secretion from the gut, adipose and pancreas. Although we previously showed that weight was stable within individuals during each domiciled calorimetry period when the diets were consumed in random order[12], we uncovered a small, clinically insignificant body weight reduction on both diets during the metabolic ward periods, and the loss was greater on the MBD than on the WD ($-625.6 \pm 196.5$ g MBD; $-134.4 \pm 156.1$ g WD; $P = 0.04$; Fig. 4a). This change in weight was accompanied by a trend towards greater loss of fat mass on the MBD than on the WD ($-289.9 \pm 97.30$ g MBD; $-64.7 \pm 84.6$ g WD; $P = 0.06$) without a change in lean mass ($-365.9 \pm 251.2$ g MBD; $-99.14 \pm 201.7$ g WD; $P = 0.45$; Fig. 4b, c). This suggests that the additional fecal energy loss on the MBD was sufficient to promote a modest change in body composition despite equivalent metabolizable energy intake based strictly on existing food digestibility paradigms. These paradigms do not account specifically for the microbial biomass or microbial energy harvest[15].

One of the gaps in prior human studies was the lack of a precise quantitation of the entire energy balance equation. In addition to our evaluation of energy intake (Supplementary Table 1, Supplementary Fig. 3b) and fecal energy loss to derive host metabolizable energy (Fig. 1a–c), we measured energy expenditure with whole room indirect calorimetry over 6 days and found no diet difference in sleep metabolic rate (in kcal/day) by diet ($P = 0.16$; Fig. 4d), despite being able to detect an a posteriori 26.5 kcal/day difference[12]. This suggests that, under conditions of fixed energy intake, the main quantitative contribution of the gut microbiome to host energy balance was through its effect on energy harvested from the diet, particularly when

sufficient substrates were available for microbial fermentation, as with the MBD.

The relationships among diet composition, gut microbes, and colonic transit time (CTT) are complex, multi-directional, and vary within individuals over time and between individuals[38]. Given the potential importance of CTT on the microbiota-driven host response to dietary manipulations, we evaluated whole-gut transit using a pH-sensing radiotransmitter device. This device has advantages to other methods (such as the use of scintigraphy or radio-opaque markers) including that it is noninvasive, generates pH, temperature and pressure data, provides whole gut and regional data, and importantly, the test is standardized to improve reliability of interindividual and longitudinal assessments[39]. In addition, the assessment was done under conditions of energy balance and with controlled diets that were customized to meet exactly the needs of each participant. This differs from other approaches that have evaluated gut microbiome-CTT interactions[40] and is an important advancement given the critical role of diet composition and quantity on both the gut microbiome and CTT. We did not find a statistically significant difference in CTT by diet ($29.7 \pm 4.4$ h on MBD vs. $39.2 \pm 6.2$ h on WD; $P = 0.14$; Fig. 4e). Gastric emptying evaluated by acetaminophen appearance in the blood after a fixed liquid meal also was not different by diet (Supplementary Fig. 6a). The pH of the colon can be an indicator of microbial fermentation activity. Neither the median pH of the entire colon (which reflects both fermentation and the impact of food mixing in the colon) nor the median pH within a 1-h window of the ileocecal passage (which is impacted primarily by microbial fermentation products)[41] differed by diet ($P = 0.11$ and 0.23, respectively; Fig. 4f; Supplementary Fig. 6b). The lack of statistically significant effects was likely due to the substantial amount of inter-individual variability in CTT, gastric emptying and colonic pH in response to each diet, confirming the complex and individualized relationships between diet and each of these parameters, which may be critical to understanding the host-microbiota axis within individuals[38].

We hypothesized that the MBD might decrease appetite relative to the WD via the inclusion of high-fiber foods and production of metabolites through gut microbial fermentation[42]. We evaluated this via subjective appetite scores (visual analog scale) and ad libitum food intake (secondary endpoints). This hypothesis was not substantiated by our data (Supplementary Fig. 6c–h). Thus, the observed negative energy balance and minor changes in body composition on the MBD

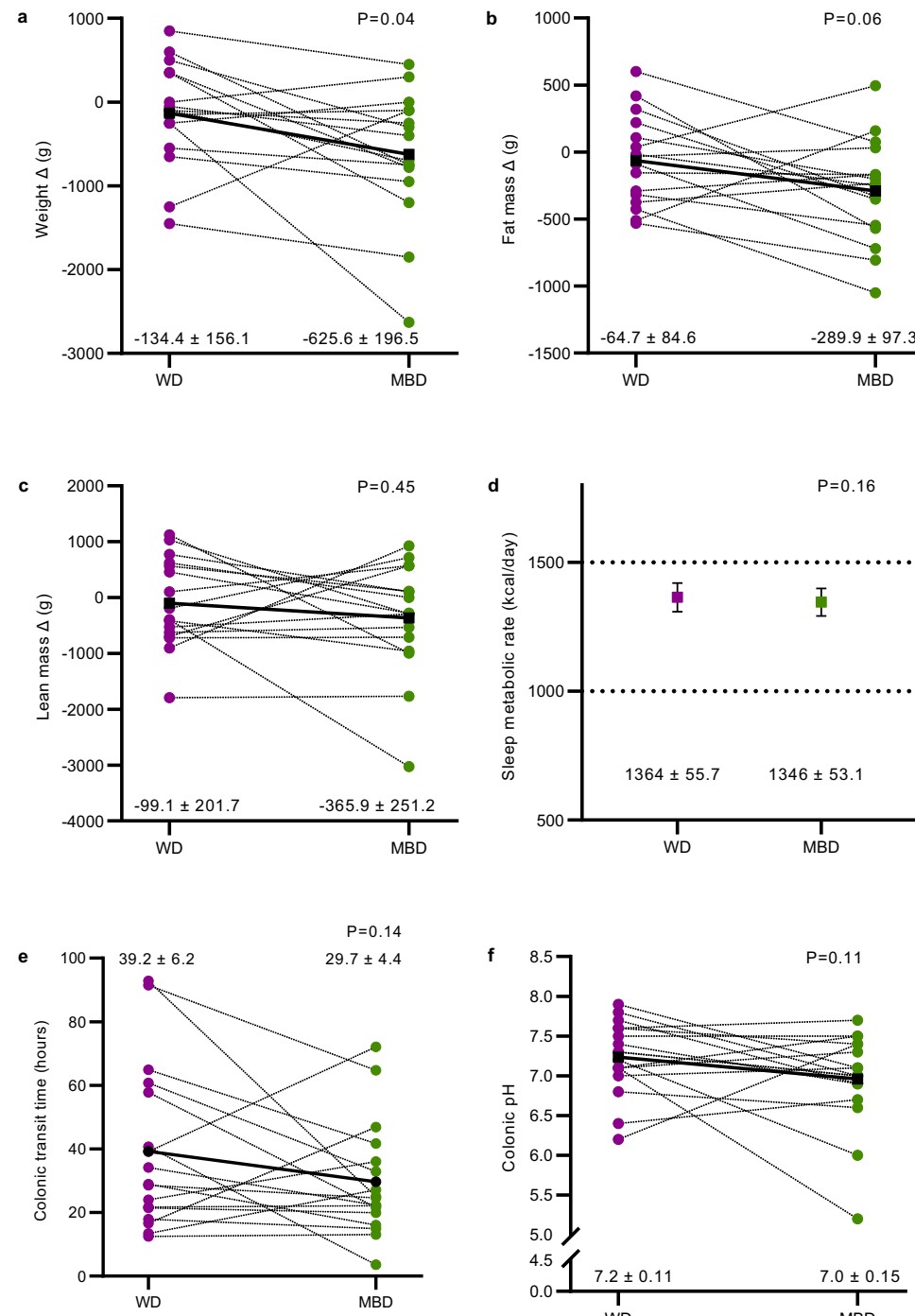

**Fig. 4 | Host energy stores and energy expenditure in response to diet-gut microbiome interactions. a–c** Weight, fat mass and lean mass changes on the WD vs. MBD; $n = 16$ per diet. **d** Energy expenditure (sleep metabolic rate extrapolated to 24-h); **e, f** Colonic transit time and median colonic pH; $n = 17$ per diet for all panels.

Error bars in panel **c** are displayed as s.e.m. *P* values are from linear mixed effects regression models and denote a statistically significant effect of diet on each endpoint. Source data are provided as a Source Data file.

did not trigger a compensatory change in appetitive behaviors or food intake compared to the WD.

The mammalian gut senses nutrients and microbial fermentation products and is part of the larger enteroendocrine system that plays a key role in maintenance of energy homeostasis[43]. Cumulative negative energy balances can result in body weight reductions. However, the regulation of body energy stores involves neural circuits in the hindbrain and hypothalamus, proximal and distal gut hormone secretions and adipose tissue neural and endocrine signals to the brain[44]. We explored several potential mechanisms by which the gut microbiome

might regulate body weight beyond the observed negative energy balance. On the second-to-last day of each domiciled period, we measured fasting and postprandial levels of several circulating hormones known to regulate appetite at 18 timepoints over 12-h (secondary endpoints). Consistent with the slight, but measurable decrease in body fat stores on the MBD, secretion of the adipose tissue hormone leptin had a significantly lower incremental area under the curve (iAUC) on the MBD ($P = 2.39 \times 10^{-5}$; Fig. 5a). A reduction in circulating leptin is known to increase food intake[45]. GLP-1 is a satiety-promoting gut incretin hormone[46] secreted by L-cells in the proximal

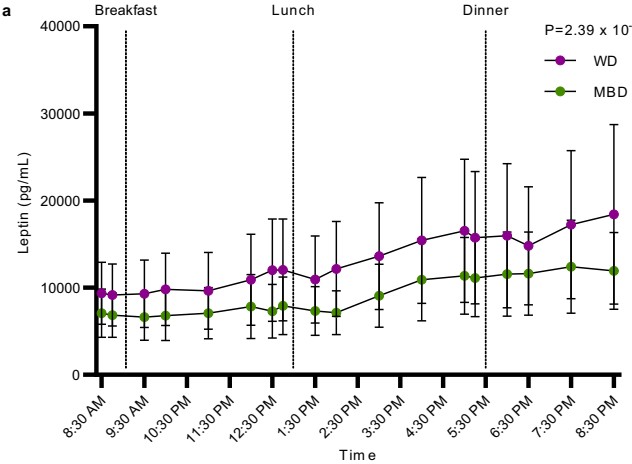

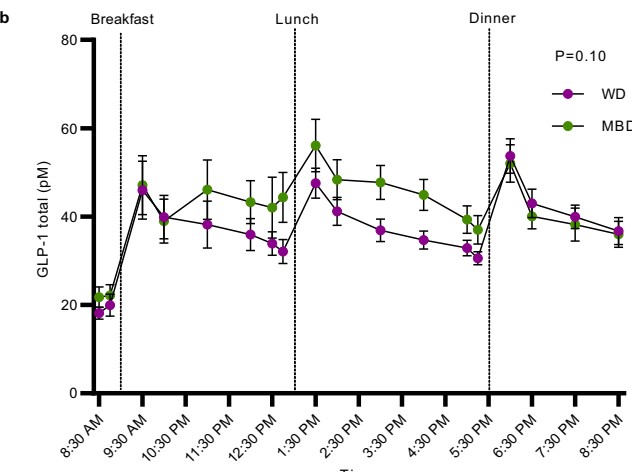

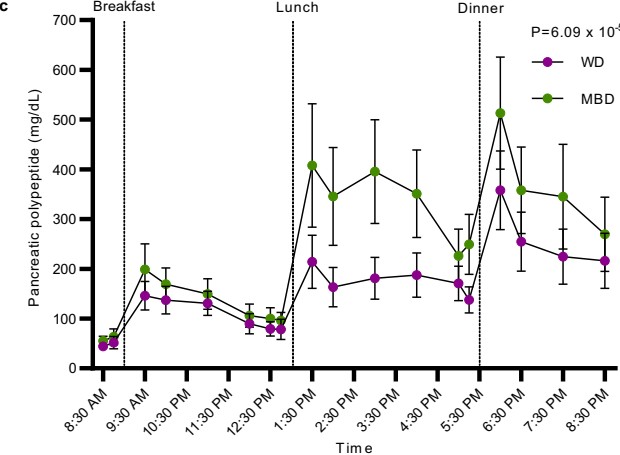

**Fig. 5 | An adipose-pancreas-gut appetite-modulating axis in response to diet.**
**a–c** Leptin, GLP-1, and pancreatic polypeptide iAUC, respectively (*N* = 15 per diet).
All data reported as mean ± s.e.m. *P* values are from linear mixed effects regression
models and denote a statistically significant effect of diet (or trend for an effect in
the case of total GLP-1) on each hormone. Source data are provided as a Source
Data file. GLP-1 Glucagon-Like Peptide 1, iAUC Incremental Area Under the Curve,
MBD Microbiome Enhancer Diet (green), WD Western Diet (purple).

gut in response to meals and from the distal colon in response to gut
microbiome metabolites including SCFA[42]. The increase in fecal and
serum SCFA on the MBD was accompanied by a trend of increased
GLP-1 iAUC (*P* = 0.1; Fig. 5b), with a significantly higher AUC at lunch
but not at breakfast or dinner (*P* = 0.009, 0.22 and 0.73, respectively)

on the MBD compared with the WD. Pancreatic Polypeptide (PP),
another satiety-promoting hormone released from the pancreas[46], had
a 1.4-fold increase in iAUC on the MBD (*P* = 6.09 ×10⁻⁵; Fig. 5c).
Therefore, the short-term negative energy balance within our experi-
mental paradigm did not trigger the compensatory food-intake
responses expected from the change in body fat and leptin. Further
experiments should pursue this hypothesis.

## Microbial contribution to human energy balance
Given the robust response to our diet intervention by the gut micro-
biome and host, we sought to determine the quantitative contribu-
tions of the gut microbiome to energy balance versus the impact
driven solely by food digestibility[10]. Host metabolizable energy on the
WD showed little interindividual variability (94.1–97.0%; Fig. 1b) since
most nutrients were absorbed in the small intestine and were inac-
cessible to the gut microbiome. However, the range of host metabo-
lizable energy in response to the MBD was much broader (84.2–96.1%;
Fig. 1b). The range translates to 73–390 non-metabolized kcals/day (vs.
59–185 kcals/day on the WD), a clinically meaningful quantitative dif-
ference that could tip the scale towards a greater negative energy
balance.

This led us to postulate that the quantitatively important varia-
bility in host energy balance could be associated with the repertoire of
gut microbes in the colon. To test this, we asked whether the variability
in host metabolizable energy on the MBD could be related to a unique
microbial signature. To identify those microbial signatures, we derived
regression coefficients describing each microbe's association with the
independent variable of host metabolizable energy using MaAsLin2's
compound Poisson regression model[26]. In total, host metabolizable
energy was associated with the relative abundance of 16 species
(Supplementary Fig. 7a, b). Four of those species had Q < 0.05 and
effect size ≥2 and have been identified as differentially abundant after
weight loss (due to bariatric surgery[47] or caloric restriction[48]) and in
bile acid metabolism[49], suggesting a potential role in weight regula-
tion. Our results were not reproducible with an alternative nonpara-
metric method (Kendall's tau-b correlation[50]; Supplementary Data 3)
and should be considered hypothesis-generating. Future studies that
are designed and powered to explore the microbial species that
explain the variability in host ME on the MBD are needed to confirm
these results.

We next embarked on a series of mathematical modeling runs[25] to
estimate the gut microbial contribution to host energy balance. We
previously reported that our in silico model estimates the dual impact
of host digestion and microbial fermentation on macronutrient uptake
in the small and large intestine and ultimately, on host metabolizable
energy. The model also estimates the amount of SCFAs absorbed by
the host due to microbial fermentation in the colon and the associated
biomass[14]. We applied an updated version of this model[25] to predict the
host metabolizable energy we measured in our study by inputting
actual energy intake components and fecal energy in grams COD/day.
Our previously published model used a fixed CTT of 48 h, which is a
reasonable population-level estimate for healthy adults[51]. With a fixed
CTT, the mean modeled host metabolizable energy for participants on
the MBD was 92.4 ± 0.001% and for WD was 95.2 ± 0.001% (Fig. 6a).
This is similar to the mean host metabolizable energy we measured on
the MBD and the WD (89.5 ± 0.73% and 95.4 ± 0.21%, respectively;
Fig. 1b). However, the model was biased as evidenced by the linear
distribution of the points which estimated essentially the same meta-
bolizable energy for each person in contrast to the variability in the
measured metabolizable energy (Fig. 6a) and Bland–Altman plot
(Supplementary Fig. 7c). We hypothesized that we could reduce the
model's bias by incorporating measured CTT since it is a key mod-
ulator of microbial composition, fermentation, and host energy
balance[38]. Incorporating the measured CTT values reduced bias based
on greater reproducibility (concordance correlation coefficient: 0.514

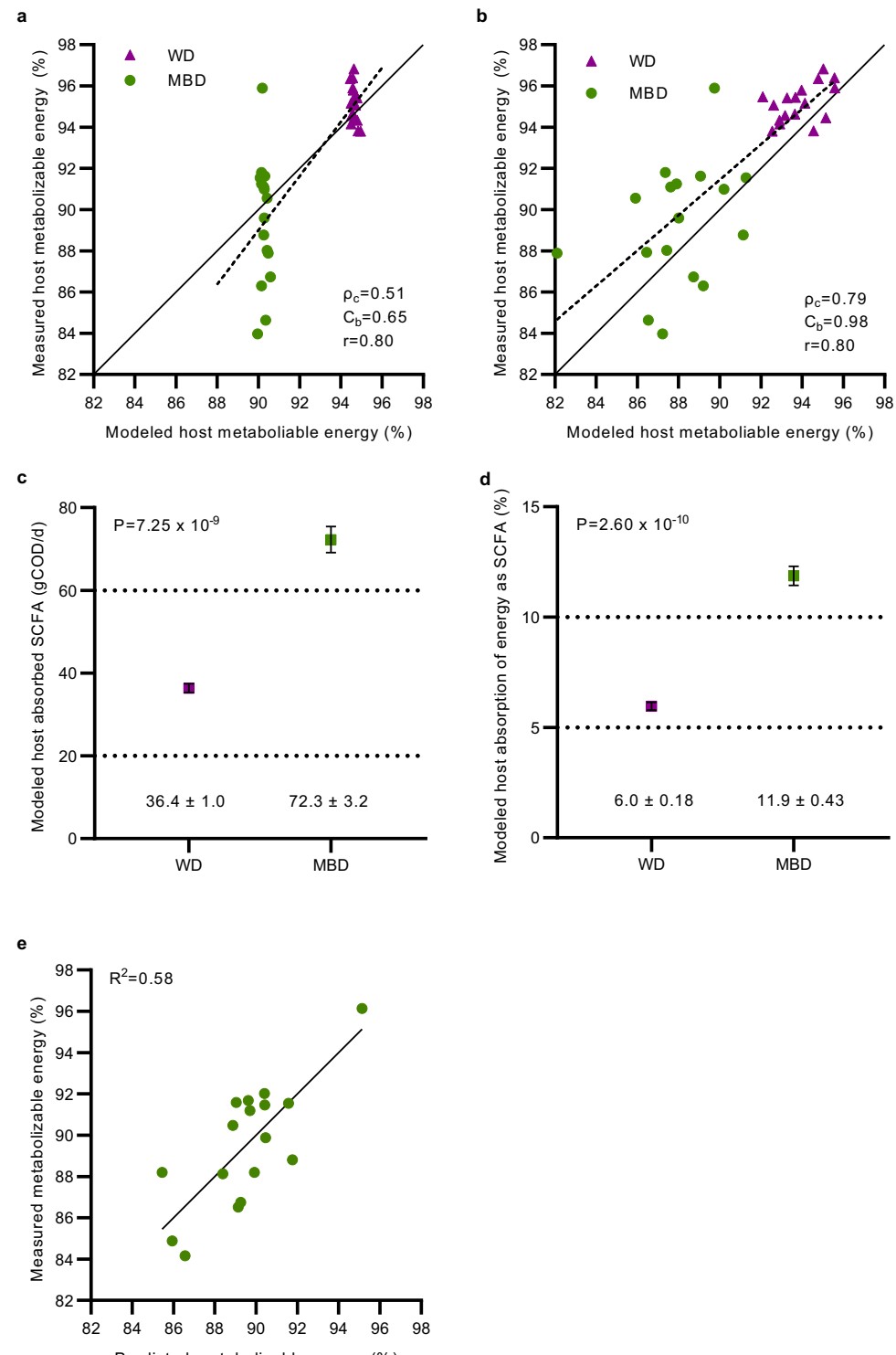

**Fig. 6 | The contributions of the gut microbiome to host metabolizable energy.**
**a** Concordance correlation coefficient plot between predicted (by modeling) and measured host metabolizable energy (ME) using the same fixed CTT (48 h) for all participants. Dashed line is a simple linear regression between pairs of data; solid line is the identity line (perfect reproducibility between measured and modeled data). **b** The same plot with each participant's measured CTT. **c** Plot shows total energy absorbed by the host as SCFAs in grams COD per day (gCOD/day) for the WD and the MBD. gCOD were calculated as the sum of acetate, propionate, n-butyrate, and iso-butyrate absorbed. **d** The percentage of COD absorbed as SCFAs adjusted for total energy intake (in gCOD/day). **e** Scatterplot of predicted and measured host ME on the MBD; predicted host ME was obtained from the model

selection procedure which estimated that 6-day fecal propionate and 16S rRNA gene copy number (a surrogate of biomass) jointly explained 58% of the variance in host ME. Thus, the R-squared for the simple linear regression of predicted and measured host ME is 0.58. $N = 17$ per diet for all panels. For panels **c** and **d**, data reported as mean with error bars showing the s.e.m. A paired samples t-test by diet was used to generate the $P$ values in panels c and d. Source data are provided as a Source Data file. ρc concordance correlation coefficient (reproducibility), Cb bias correction factor (accuracy), COD Chemical Oxygen Demand, CTT Colonic Transit Time, Host ME Host Metabolizable Energy, IQR Interquartile Range, MBD Micro-biome Enhancer Diet (green), SCFA short-chain fatty acids, r Pearson's correlation coefficient (precision), RA Relative Abundance, WD Western Diet (purple).

with fixed CTT and 0.789 with measured CTT) and accuracy (bias correction factor: 0.645 with fixed CTT and 0.983 with measured CTT; Fig. 6a, b). Furthermore, systematic and proportional biases were minimized, as shown by a Bland–Altman plot (Supplementary Fig. 7d). Collectively, these data suggest that CTT is an important factor for estimating host metabolizable energy.

High-fiber diets should increase absorption of SCFAs due to colonic microbial fermentation of fiber and resistant starch[52]. Our model predicted that more total energy (g COD) as SCFAs was absorbed by the host on the MBD, compared to the WD ($72.3 \pm 3.2$ gCOD/day of microbially-derived SCFAs on the MBD vs. $36.4 \pm 1.02$ gCOD/day on the WD; $P = 7.25 \times 10^{-9}$; Fig. 6c). When we adjusted the SCFA absorption for energy intake and calculated a percentage, we found a nearly 2-fold greater absorption of energy as SCFAs on the MBD as compared to the WD ($P = 2.60 \times 10^{-10}$; Fig. 6d). Therefore, despite less total energy being absorbed by the host on the MBD, a larger proportion was derived from SCFAs. Consistent with our experimental data, our model strongly supports a significant microbial contribution to host metabolizable energy and, therefore, the overall energy balance.

The wide interindividual variability in host metabolizable energy in response to the MBD is likely related to a combination of host and microbial factors. We postulated that we would be able to identify key parameters within our study sample that explain a portion of this variability. Thus, we undertook an exploratory, multi-step statistical process to identify potential host and microbial determinants of host metabolizable energy using data from the MBD only. We did so with an aim to consider a limited number of factors given our sample size. Following consideration of 15 potential factors as described in Methods, the final step in this process was a multivariate selection procedure into which one host factor (CTT) and two microbial factors (fecal propionate and biomass) were entered. The selection procedure chose fecal propionate and biomass—both microbial factors—for the final optimized model and revealed that these two variables jointly explain 58% of the variance in host metabolizable energy. According to the final model, each standard deviation (SD) increment in 6-day fecal propionate (858.6 mg) was associated with a 2.1% lower host metabolizable energy (95% CI 0.95, 3.2), while each SD increase in biomass (0.49 log of 16S rRNA gene copy number) was associated with a 1.6% lower host metabolizable energy (95% CI 0.44, 2.7) (Fig. 6e).

## Discussion

Microbial communities in the gut have a profound impact on mammalian host endocrinology, physiology, and energy balance, with most causal inferences historically restricted to preclinical animal models[5,7,8]. Prior human studies exploring the relationships among the gut microbiome, obesity and energy balance lacked the deep phenotyping, precise methodologies, and rigorous controls that are instrumental for drawing causal inferences with respect to human health. Our central finding was that a diet designed to feed and modulate the colonic gut microbiome, under conditions of fixed energy intake and physical activity, led to reduced metabolizable energy to the host due to increased fecal energy output consisting of undigested food, 16S rRNA gene copy number (a surrogate of fecal bacterial biomass), and microbial metabolites but not to changes in energy expenditure. Thus, the greater fecal energy loss on the MBD was not just due to undigested food, but also to an in increase in fermenting gut microbes and their metabolites. Although higher energy harvest by microbes is believed to lead to more energy being absorbed by the host based primarily on preclinical models[8], our results show the opposite: host metabolizable energy was lower due to higher fecal energy loss on the MBD.

The greater fecal energy loss translated to an additional 116 kcal/day lost in feces when participants were fed the MBD as compared to the WD. The clinical significance of this difference can be inferred from the reduction in food intake needed to maintain a weight-reduced

state with obesity pharmacotherapy which has been modeled to be approximately 200 kcal/day[53]. In addition, the cumulative impact over time of a 116 kcal/day energy deficit is in alignment with the population-level impact of "small changes" in energy balance to promote weight loss[54].

The direction of change in energy absorbed from the diet by the host (lower on the MBD) was consistent in 16 out of 17 study participants. This means that the gut microbiome of most of our participants had the capacity to utilize available dietary substrates as evidenced by changes in relative abundance of species capable of utilizing those substrates. This observation is contrary to the "extinction" hypothesis proposed in mice[55] and suggests that, with rigorously controlled dietary conditions that vary markedly in the amount of substrate delivered to the colon with a comparable kcal and total macronutrient profile, healthy humans harbor microbiomes which are adaptable and/or have sufficient functional redundancy to overcome certain extinctions that might be imposed by diet or other factors.

The reduction in host metabolizable energy on the MBD relative to the WD was not accompanied by a change in energy expenditure or an increase in hunger or ad libitum energy intake. However, the significant diet-induced modulation of the gut microbiome was accompanied by a modest change in weight/body composition and robust enteroendocrine signals from the adipose-pancreas-gut axis. Our results support the hypothesis that an intentional remodeling of the gut microbiome through provision of adequate dietary fiber, resistant starch, and a focus on whole, minimally processed foods resets the integrated sensing mechanisms known to affect food intake and body energy stores. One or more of these mechanisms or other unknown mechanisms might be responsible for the associations between a diverse human gut microbiome and lower body mass index in free-living humans[5]. Future host-diet-gut microbiome research should delve into the complex and interrelated systems that control body weight.

The quantitative contributions of gut microbes to host energy balance were addressed in two ways. First, the microbial biomass was modeled to contribute to >25% of the total fecal energy on both diets. Second, fermentation increased as evidenced by increased fecal and serum SCFAs on the MBD as compared to the WD. Thus, host energy absorption shifted towards microbially produced SCFAs and away from proximally digested and absorbed nutrients. While the quantitative contribution of microbially generated SCFAs as inputs to host energy balance was negated by the additional loss of microbial biomass in the feces, the uptake of more microbially produced SCFAs was associated with increased total GLP-1 and PP concentrations which may trigger important energy homeostasis signaling cascades to promote satiety and suppress hunger[56,57].

We also found a taxonomic signature that was in alignment with the expected impacts of the substrates available to the gut microbes on the two diets. Many of the species detected at higher abundance on the MBD were fiber degraders and/or butyrate producers. We posit that the higher relative abundance of microbes that produce SCFAs could modulate several components of the energy balance equation. For example, *Lachnospira pectinoschiza*, *Eubacterium eligans*, and likely the uncharacterized *Oscillibacter* are butyrate producers[28–30]. Butyrate plays important roles in host energy balance by stimulating the release of satiety hormones such as GLP-1[58] and accelerating CTT[59]. In addition, acetate stimulates the release of satiety hormones[60] and acts as a satiety signal[61].

Host metabolizable energy was highly variable on the MBD. Given our tight control of energy intake and energy expenditure, this suggests that the microbial contribution to this variability was greater in some hosts than others. Indeed, with a proportionally equivalent input of substrates for microbes, fecal energy losses varied over an ~5-fold range. Next steps should include investigating the mechanisms by which the microbial communities in the human colon modulate energy

balance and their interaction with host factors such as CTT, which will provide valuable quantitative data to drive personalized strategies to optimize host-microbiota-diet interactions and prevent or treat obesity.

Our in silico mathematical model, which accurately estimated host metabolizable energy measured in our study, allowed us to predict the quantitative contribution of biomass to fecal energy harvest and the 2-fold greater uptake of microbial derived SCFAs on the MBD than on the WD. In addition, we found that microbial biomass and fecal propionate (a surrogate of colonic fermentation) explained over half of the variance in host metabolizable energy. This further supports our hypothesis that the gut microbiome is an important component of diet modulation of host energy balance. We believe that by optimizing CTT, biomass, and SCFA production through diet, among other factors that may be revealed in future studies, the highly adaptable gut microbiome can serve as a target for personalized medicine[62].

Our results collectively indicate that when dietary substrates are less available to the gut microbes (as with the WD), the microbes are "starved" of host diet-derived substrates. This is in agreement with the findings from Sonnenberg et al.[13]. The lower 16S rRNA gene copy number on the WD suggests a decrease in microbial biomass due in part to lower fermentable substrate availability to the microbes from the diet. Lower fecal and serum SCFAs on the WD point to lower microbial fermentation, which is an indicator of reduced microbial energy harvest[8]. In addition, the increased relative abundance of mucin degraders on the WD suggests that the microbiota was "starved" of diet-derived substrates and turned to the host-derived energy sources such as mucin. Mouse models have shown that *B. thetaiotaomicron*, which normally degrades glycans from plant-based foods, consumes host-derived mucin when diet-derived glycans are not available in sufficient quantitites[63]. Similarly, mucin-degrading bacteria are more abundant in humans on calorie restricted diets or suffering from anorexia nervosa[64].

Our study had several limitations that should be considered when interpreting our results. Although our data revealed key gut microbiome contributions to human energy balance, we were unable to deconvolute the complex human host-diet-gut microbiome interactions and therefore, cannot establish whether the changes in energy balance we observed are causally attributable to the diet, the microbes, or some combination. Future directions to address these gaps include implementing bioinformatic pipelines that allow for absolute quantification of microbial species[65] and the proportion of fecal energy contributed by the gut microbes (vs. undigested food). Additional mechanistic experiments are needed in preclinical models and bioreactors to establish the specific physiological pathways driven by communities of microbes[66], identify systemic lipids, metabolites or proteins that mediate diet-host-microbe interactions, and understand how microbes utilize dietary components[67]. Given the small sample size of our precisely controlled study, selection bias may limit generalizability to other populations. Future studies should confirm and expand our findings in larger study samples. Nonetheless, the randomized crossover design vastly reduces the likelihood of confounder bias and yields outstanding internal validity. Larger studies could enable subgroup analyses to inform whether effect sizes vary by sex or other participant characteristics.

A key open question to advance this field is whether and how obesity or caloric restriction impact diet-gut microbiome effects on human energy balance. Given the size and scope of the global obesity epidemic and its continued increase, new solutions are needed. The scientific community has recently reoriented itself towards population interventions that promote small changes in energy intake and expenditure as a means of preventing weight gain[54]. This study demonstrates the potential to enact the "small changes"[54] principle through the consumption of whole foods to modulate the gut microbiome. Such a simple principle could be a useful population-level tool

to fight the global obesity epidemic. Future experiments should focus on the microbial or host mechanisms that underly the observed large inter-individual variability in the response to delivery of greater dietary substrates to the gut microbes. These mechanisms can then be targeted with precision nutrition approaches.

## Methods
The details of the design of this trial (NCT02939703) have been previously reported[12]. We summarize key elements below and include details on elements not reported elsewhere. The full study protocol is provided in Supplementary Data 4.

### Study participants
The results presented are from a clinical trial conducted in compliance with all applicable ethical and institutional research requirements. The study was approved by the AdventHealth Institutional Review board (Orlando, FL, USA). All participants provided informed consent. Remuneration was done in accordance with local norms and with the approval of our Association for the Accreditation of Human Research Protection Programs, Inc. (AAHRPP) Accredited IRB to ensure we provided fair compensation without inducement. There were no major changes to the methods after the trial commenced. We made two minor changes: 1) clarified the depression exclusion to provide a clear timeline for diagnosis and severity parameters warranting exclusion; 2) added an anemia exclusionary criterion out of an abundance of caution due to the multiple blood draws at the end each diet period. We recruited approximately equal numbers of males and females 18–45 years of age with a BMI ≤ 30 kg/m$^2$ who were weight stable, otherwise healthy, and had not used antibiotics for the 3 months prior to screening between June of 2017 and August of 2019[12]. The last participant visit occurred in October of 2019. Adverse events were monitored at each contact with the participant and reported according to Institutional Review Board guidelines.

### Design overview
This was a randomized crossover study with a control Western Diet (WD) compared to a Microbiome Enhancer Diet (MBD) where each participant served as their own control, thereby minimizing the impact of confounders[68]. We applied block randomization stratified by sex. The randomization code was generated by the study statistician who worked directly with the study dietitian in charge of assigning menus to participants. Participants were enrolled by the study coordinator. In order to balance sex, we randomly assigned 3 blocks to each sex. Within each block ($n = 6$), participants were randomly assigned using simple randomization to one of two diet sequences with a 1:1 allocation ratio using SAS PROC PLAN. Eight participants were randomized to sequence 1 (WD followed by MBD) and 9 participants were randomized to sequence 2 (MBD followed by WD).

After an initial assessment period to establish outpatient dietary intake requirements (Days 1–9), all food was provided to participants, and they consumed the meals outpatient for 11 days (Days 10–20 and 39–49) and inpatient for 12 days (Days 21–32 and 50–61). Included was a minimum 14-day washout between diet periods. During the 14-day washout, participants returned home and were instructed to resume their usual diet and physical activity. The following additional restrictions were implemented prior to each study period: avoidance of alcohol during outpatient feeding periods, no caffeine up to 72 h before admission, no artificial sweeteners, and no strenuous physical activity 48 h prior to admission. All endpoint assessments were conducted while participants were housed in our metabolic ward[12].

### Armband accelerometry
An armband accelerometer was placed to measure activity and estimate free-living energy expenditure (SenseWear Pro 3 Armband, BodyMedia Inc.) as well as free-living sleep duration, as previously

described[12]. These data were used to estimate outpatient calorie requirements.

## Clinical assessments

Health status was determined by medical history, physical examination, standard blood chemistries (AdventHealth Laboratory, Orlando, FL, USA), and the Bristol Stool Scale to evaluate stool type based on shape and consistency, with scores of 3-4 indicating neither constipation nor loose stools[17].

## Whole-room indirect calorimetry

Energy expenditure and all its subcomponents was evaluated every 24-hours with whole room calorimetry in two 6-day blocks per diet (days 24–29 and 53–58[12]) following published standards of operation[69]. We present the results of sleep metabolic rate (kcal/24-h) because it has the lowest interindividual variability. Activity was tightly controlled during the day to maintain spontaneous physical activity consistent within and between participants and to ensure consistent times of meals, exercise, type of activity, rest and sleep[12]. Motion-free sleep was calculated from the radar motion detector in the calorimeter by removing all minutes with ≥6 counts of movement. This is a surrogate of high-quality sleep that we use to minimize the effects of small amounts of involuntary motion during sleep on sleep energy expenditure[20]. Supplementary Table 2 demonstrates the calorimetry schedule.

## Energy balance

Energy balance was estimated by subtracting estimated metabolizable energy intake (calculated by menu design software based on actual food intake) from energy expenditure measured by whole room calorimetry[70].

## Host metabolizable energy

To calculate host metabolizable energy (primary endpoint), we converted energy intake in kcals to grams COD using our published model[24]. That allowed us to compute the percent of energy metabolized by the host after accounting for fecal energy loss which was also measured in COD. To relate this percentage back to kcals and determine the number of daily kcals that were not absorbed by the host, we multiplied host metabolizable energy percentage by energy intake in kcals.

## Dietary intervention

Our study diets were designed to maximize the differences of dietary substrate availability to gut microbes with the MBD while minimizing it with the WD. To achieve this, the MBD was higher in fiber and resistant starch, which are known substrates for microbial fermentation[13]. We also provided larger food particles (whole nuts vs. nut butter, for example) because fine grinding of foods makes nutrients more bioavailable to the human host and thus, less available to the gut microbes[71,72]. One final element of our diets was minimizing processed foods on the MBD, in contrast to the known excess of processed foods in the WD. Accumulating evidence indicates that processed foods, in addition to lacking fiber and having smaller particle sizes, negatively impact host health in part via the gut microbiome[73]. Details of the diet design, including sample menus can be found in our trial design publication[12].

Diets were prepared in our metabolic kitchen based on kcals needed to maintain energy balance as determined by whole-room indirect calorimetry. Diets were designed with menu software (Pro-Nutra Version 3.5, Viocare, Inc, Princeton, NJ) that proportionately calculated diets based on each participant's energy needs. Duplicate meals were prepared during all calorimetry days and evaluated for energy content as a quality-control step (Eurofins, Madison, WI). Nutritional composition of the diets was based on the menu software

database (USDA Database Standard ref. 23), with the exception of resistant starch, because it is absent from all currently available nutritional databases. We limited foods containing resistant starch on the WD and then estimated the content on both diets based on published estimates of resistant starch content of common foods[74]. Diets were equivalent in metabolizable energy and proportions of macronutrients. As much as possible, we used similar types of foods on both diets to minimize differences in micronutrients. Supplementary Table 1 shows the energy, macronutrient, and drivers in each diet. Consumption of 100% of provided foods was required. Diet adherence was monitored during the 11-day outpatient phase at clinic visits 2 or 3 times per week where at least one meal was consumed on site. During the domiciled metabolic ward phases, all meals were monitored. Adherence was calculated by weighing back uneaten food (if any) and recalculating dietary intake as a percentage of provided diet vs. unconsumed diet[12]. Sample menus are available as supplementary material in our trial design publication[12].

## Free-living dietary intake

Outpatient dietary intake was evaluated using the validated food frequency questionnaire, the Diet History Questionnaire II (Diet History Questionnaire, Version 2.0. National Institutes of Health, Epidemiology and Genomics Research Program, National Cancer Institute, 2010), on which patients self-reported the frequency and portion sizes of food items consumed over the past 12 months. Diet Calc software was used to estimate nutrients consumed based upon the United States Department of Agriculture food database (Diet*Calc Analysis Program, Version 1.5. National Cancer Institute, Applied Research Program).

## Sample collection, processing and shipment

Blood, urine, and fecal samples were collected in our metabolic unit using standard protocols (Translational Research Institute, Orlando, FL, USA). Fecal samples were collected each time they were produced during the 6-day measurement periods in the whole room calorimeter. Sample weights were tracked upon collection and as aliquots were prepared. Fecal samples were processed within an hour of production under an anaerobic hood and were maintained on ice during processing. After mixing with a sterile spatula, samples were sub-aliquoted for various downstream applications. Samples for metagenomic sequencing and SCFAs were snap frozen without additives and stored at −80 °C. They were shipped overnight on dry ice. Any fecal sample not needed for method-specific aliquots were stored (sealed) in the original collection container at −20 °C within 60 min of collection. At the end of each 6-day calorimetry period, all collection containers were opened, and all frozen samples were transferred into a single, large, homogenization container to create a composite sample (without additives) that was used to measure fecal energy and biomass. The composite was partially thawed on ice while remaining sealed and then homogenized, on ice, using a sterile paddle homogenizer. The composite sample was stored at −80 °C until used or shipped overnight on dry ice[12].

## Weight and body composition

Weight (fasting and in a gown) was measured daily during the 12-day metabolic ward stay on a calibrated scale. Body composition was assessed with dual energy x-ray absorptiometry the day prior to entering the calorimeter (Days 23 and 52) and after exiting the calorimeter (Days 31 and 60) with a two-day window allowed for the pre or post measurement.

## Fecal energy

Fecal energy was measured with chemical oxygen demand (COD) at the AdventHealth Translational Research Institute (Orlando, FL, USA) as per our previous publication[24]. Briefly, COD was measured per manufacturer's protocol using a reactor digestion method with high-

range digestion vials followed by a colorimetric assay (HACH, Loveland, CO; Product # 2125925). To ensure that fecal energy was accurately reflective of 24-hour fecal production, we utilized the non-absorbable, non-digestible fecal marker polyethylene glycol (PEG). Participants consumed 1.5 g/day (0.5 g/meal) of PEG of molecular weight 3350 g/mol (PEG3350). The PEG3350 was procured by a compounding pharmacy that prepared 0.5 g capsules (percent error = 2.8%) (Pharmacy Specialists, Altamonte Springs, FL). The details of the PEG assay are below. Fecal energy was measured in 6-day composites of feces collected in our calorimeters. We normalized fecal energy produced to the weight of all feces produced in those 6-days and then to PEG recovery. Fecal energy loss was converted to host metabolizable energy by calculating the percentage of energy that was lost in feces (in g COD) relative to total energy intake (in g COD). The conversion from energy in COD to kcals lost in feces per day (non-metabolizable kcals) was calculated by multiplying total EI in kcals by the percent host metabolizable energy.

### Polyethylene glycol assay

We utilized a method that is slightly modified from the initial published method by Sadilek et al.[75]. Key modifications include quantitation based on the +2 charged PEG3350 polymers instead of the +4 charged polymers and the inclusion of an internal standard. Sample preparation was also slightly modified. Briefly, samples were prepared by a 1:1 dilution with Nanopure water and homogenized. Two grams of sample was diluted in 14 ml Nanopure water that included a final concentration of 1.5 uM internal standard (monodispersed PEG, MW 2160 g/mol; Quanta Biodesign, Plain City, OH; Product # 10897). An HPLC-MS method was used for the separation and detection of PEG3350 in human fecal samples[75]. The modfied assay was transferred to ARL Biopharma for subsequent PEG quantification on study samples (Oklahoma City, OK). The assay is linear as evidenced by the R² of the calibration curve (0.9987). The linear range of the assay was from 0.1 uM to 20 uM with PEG3350 recovery ranging from 96.2 to 104.5%. The relative standard deviation of the assay was 1.8%. There was no co-elution of analyte with expected excipients or related compounds in chromatograms demonstrating the assay is specific for PEG3350.

### Quantification of bacterial 16S rRNA genes

Quantitative PCR (qPCR) was performed (Arizona State University, Tempe, AZ, USA) with triplicate PCR reactions as previously described[76] in a Thermofisher Applied Biosystems Quant Studio 3. Universal primers 926F (5′ – AAACTCAAAKGAATTGACGG – 3′) and 1062R (5′ - CTCACRRCACGAGCTGAC – 3′) were used. Calibration curves using 7 data points were generated on each run using plasmids with 16S rRNA genes, and adding a plasmid concentration to achieve copy numbers in the range from $10^1$ to $10^9$ per reaction. Reaction mixtures with a final volume of 20 μL, comprised of 10 μL 2× Fast-Start SYBR green, 0.6 μL each forward and reverse primer (final concentration, 0.3 μM), 2 μL DNA template (equilibrated to 10 ng), and deionized H₂O to 20 μL. Themocycler conditions were 95 °C for 5 min, followed by 30 cycles of 95 °C for 15 s, 61.5 °C for 15 s, and 72 °C for 20 s, and a final elongation step at 72 °C for 5 min. Standards were made by cloning the *E. coli* 16S rRNA gene using the ThermoFisher TOPO TA Cloning Kit. Plasmids were purified using the Qiagen QIAprep Spin Miniprep Kit. Purified plasmids were quantified by Qubit. Plasmid copy number was then calculated using the following formula:

$$Copy\,Number = \frac{\left[DNA\left(\frac{ng}{\mu L}\right)\right]*6.022*10^{23}}{Plasmid\,length\,(bp)*10^9*660} \quad (1)$$

16S rRNA gene copy numbers per gram of feces were used to calculate daily copy numbers by multiplying by fecal weight and adjusting to PEG recovery.

### DNA sequencing

Fecal sample processing, nucleic acid extraction, library preparation, and whole genome shotgun sequencing were performed at the University of North Carolina at Chapel Hill Microbiome Core (Chapel Hill, NC, USA), which is supported by the following grants: Gastrointestinal Biology and Disease (CGIBD P30 DK034987) and the UNC Nutrition Obesity Research Center (NORC P30 DK056350). DNA was extracted using the QIAamp Fast DNA Stool Mini Kit and library was prepared using the Swift 2S Turbo DNA library kit. DNA was sequenced on the Illumina HiSeq 4000 PE 150 platform. Positive controls included ZymoBIOMICS Microbial Community Standard (Cat. No. D6300), Microbial Community DNA standard (Cat. No. D6305), and Gut Microbiome Standard (Cat. No. D6331). DNA-free deionized water was used as a negative control and to detect possible contamination. To avoid batch effects, fecal samples were randomized prior to nucleic acid extraction and all samples were sequenced at the same time. Mean total reads were 18,339,758, with similar read depth on each diet (19,475,004 for the WD and 17,204,513 for the MBD).

### DNA sequence processing

DNA sequencing output was quality controlled with FastQC (Version 0.12.0)[77]. Adapters were trimmed using TrimGalore (Version 0.6.5)[78]. DNA sequences were aligned to Hg38 (GRCh38.p14) using bowtie2 (Version 2.4.5)[79]. Sequences were filtered in the alignment step of sequence processing. Reads were paired. Count data was used as input into software packages for analysis. Software packages then used total sum scaling (TSS) to calculate relative abundance. DNA sequences were then analyzed for taxonomic composition with MetaPhlAn3 (Version 3.0.14)[80], using standard parameters.

### Species alpha- and beta-diversity

All calculations and analyses were conducted in R (Version 4.2.2)[81]. Taxonomic composition output from MetaPhlAn3 (Version 3.0.14) was processed for beta-diversity analysis using the "phyloseq" R package (Version 1.42.0)[82]. A rarefaction curve was created using the "vegan" R package (Version 2.6-40)[83] to determine the optimal count-depth for rarefaction. Once the optimal count-depth was determined, rarefaction was performed using phyloseq (Version 1.42.0). Alpha-diversity metrics were calculated using the "microbiome" R package (Version 1.20.0)[84]. After samples were rarified, each sample had 3,578,445 sequences. Bray–Curtis and Jaccard distance matrices were calculated on the rarefied count data using vegan (Version 2.6-40). The distance matrices were tested for significance by PERMANOVA using vegan (Version 2.6-40). Diet was the only significant term for both metrics (Bray–Curtis: $P = 0.017$, Jaccard: $P = 0.016$). Beta-dispersion was calculated, and the results tested for significance with the ANOVA-like permutation test and Tukey's HSD in vegan (Version 2.6-40). Constrained Analysis of Principal Coordinates (CAP) ordination was performed with vegan (Version 2.6-40). CAP is a multivariate linear method[85] that we used to assess how much of the variation in the beta-diversity could be explained by diet. Consistent with the linear mixed model approach used for analyzing the clinical data, diet, sequence, and period were fixed, and participant was a random factor. Statistical significance testing was performed with PERMANOVA in base R. Beta-diversity ordination figures were created using the "ggplot2" R package (Version 3.4.1)[86]. Differential abundance heatmap figures were created using the "ComplexHeatmap" R package (Version 2.14.0)[87].

### Differential abundance

Differential abundance testing by diet and associations of species relative abundance with host metabolizable energy were carried out using the output of MetaPhlAn3 in the "MaAsLin2" R package (Version 1.12.0)[26]. Taxonomic counts were filtered with a 25% prevalence cut-off. Compound Poisson multivariate linear models were used to account for zero-inflated data[26]. For differential abundance testing, diet,

sequence, and period were fixed, with participant as the random factor. For analyzing associations of species relative abundance with host metabolizable energy, metabolizable energy was the fixed independent factor, because the analysis was restricted to samples in the MBD only.

For rigor and reproducibility, we repeated the differential abundance analyses with a different method using the same parameters as the original analyses. We validated our results related to differential abundance by diet with ANCOM-BC, an approach often used to evaluate differential abundance that has been found to produce results that frequently overlap with other approaches[37]. For validating correlations between relative abundance and metabolizable energy on the MBD only, we employed Kendall's tau-b correlation coefficient[50] because it is a non-parametric test that is often used to explore the relationship between relative abundance and continuous or ordinal variables[88].

### Short-chain fatty acids
A targeted SCFA panel including acetate, propionate and butyrate was conducted for both fecal and serum SCFA (Metabolon, Inc., Mooresville, NC). For fecal SCFAs, the concentrations were adjusted for total feces produced and PEG recovery to calculate the total fecal SCFAs over the 6 inpatient calorimetry days. Acetate, butyrate, and propionate were summed to calculate total fecal SCFA and total serum SCFA.

### Appetite
Subjective ratings of appetite (secondary endpoint) were determined using visual analog scales (VAS) administered at −30, −15, +30, +60, +120, and +180 min pre/post each meal. Breakfast was fixed at 500 kcals and lunch and dinner provided 1.5 X the energy content of each participant's energy balanced diet consumed while in the whole room calorimeter, which is equivalent to 1.3X the energy needed in free-living conditions on our metabolic ward. Ad libitum intake (secondary endpoint) was allowed at lunch and dinner for assessment of changes in food intake[12]. The trapezoidal rule was used to calculate the iAUC per meal and diet for each appetite scale[89].

### Gut transit time
A radiotransmitter motility capsule was used to determine transit time and pH in the colon (SmartPill™; Medtronic, Minneapolis, MN)[12,90]. The SmartPill™ was administered while participants were in the whole-room calorimeter under a standardized protocol. It was consumed immediately after breakfast.

### Gastric emptying
Gastric emptying was assessed via acetaminophen appearance in serum after a test meal. Acetaminophen (1,500 mg) was administered at nominal timepoint zero[12]. The measurement assay was performed at the Pennington Biomedical Research Center (Baton Rouge, LA, USA).

### Enteroendocrine hormones
Enteroendocrine hormones (secondary endpoint) in plasma were evaluated after a test meal (Boost Plus or equivalent, 500 kcal) and lunch/dinner from their assigned diet at nominal timepoints −30, −15, +30, +60, + 120, and +180 min pre/post each meal[12] (Translational Research Institute; Orlando, FL, USA). GLP-1 (active), Leptin, and Pancreatic Polypeptide were measured with V-PLEX Metabolic Panel 1 Human Kit (MesoScale Diagnostics, Rockville, MD; Product # K15325D). For enteroendocrine hormones, the iAUC for the total time of measurement (-11 h) was calculated by diet. The trapezoidal rule was used to calculate the iAUC[91].

### Mathematical modeling
Previously, we developed an in silico multicompartment transit, reaction, and absorption model with these 3 compartments: upper gastrointestinal tract, lower gastrointestinal tract, and the remaining human body[14]. The model estimates human dietary absorption for the general population and humans who had sections of small intestines and large intestines surgically removed. Specifically, the model calculates the host absorption of carbohydrates, protein, and fat in the upper gastrointestinal tract and microbe-derived SCFAs in the lower gastrointestinal tract[14]. For each participant, we had daily and cumulative values for grams of carbohydrates, proteins, fat, total fiber, and resistant starch consumed based on our designed menus. To use this information in our mathematical model, we systematically converted the measurements into gCOD/day of a) Available Sugar and Starch; CHO (g) - Resistant Starch (g) – fibers (g), b) Resistant Starch (RS), c) Non-Starch Polysaccharides (NSP), d) Proteins, and e) Fat. These data were input into the model to estimate host metabolizable energy and compare it to our measured data. We then improved the model by using the measured CTT and evaluated the impact of this change by comparing actual versus modeled data. The performance of the model for estimating host metabolizable energy was evaluated by analyzing concordance correlation coefficient components: bias correction factor (Cb)- accuracy; Pearson's correlation coefficient (r) between measured and modeled metabolizable energy- precision; concordance coefficient correlation (ρc)- precision[92]. We evaluated systematic and proportional bias with a Bland–Altman plot[93]. We compared the absolute and proportional SCFA absorption with a paired two-tailed t-test. The model code with revisions specific to this manuscript can be found here: https://zenodo.org/badge/latestdoi/634925145[25].

### Statistics and reproducibility
Details of the sample size determination have been previously published[12]. Briefly, our prior modeling experiments predicted a difference of 110 kcal of fecal energy loss (COD g/day) on the MBD vs. WD for a person consuming 2000 kcal/day. Our power analysis was based on prior data on repeated measures of 24-h and sleeping energy expenditure[94] and prior COD measurements where replicate variability was known. Our sample size was adequate to detect a 120 kcal/day difference in energy expenditure and an 80 kcal/day difference in fecal COD. This small sample size was made possible by the repeated measures, precision of our measurements, crossover design, and tight control of diet and environment.

Descriptive statistics for continuous variables are presented as mean ± standard error of the mean if normally distributed or as median (interquartile range) if non-normally distributed; categorical variables are shown as counts and percentages.

Appropriate to our randomized crossover design, we used a linear mixed model (SAS PROC MIXED) with diet, period, and sequence as fixed effects and participant as a random effect to compare differences by diet in our primary endpoint (host metabolizable energy: fecal energy loss adjusted to energy intake) and most other secondary and exploratory endpoints. When the distribution of the model residuals was found to deviate considerably from normality, a logarithmic transformation was applied. For each endpoint, we included only participants with complete data for both diet interventions when the data were considered to be missing not at random. Otherwise, no data were excluded from the analyses. Three values that were considered to be missing at random for the enteroendocrine hormone data (one out of 18 serial timepoints over the day was missing for each of three participants due to temporary issues with blood draw or laboratory analysis, but not because the entire sample was missing) were imputed using the interpolation method (i.e., averaging the previous and subsequent values)[95]. We imputed the last (+180) timepoint of 6 serial timepoints for one participant for the gastric emptying assay using the last observation carried forward approach, given the relative stability of the +120 min and +180 min timepoints (median difference 0.55 ng/mL; IQR −0.55, 1.55)[95].

To identify host and microbial determinants of host metabolizable energy, we first constructed a correlation matrix with hypothesis-driven host and microbial factors that might influence the efficiency of dietary energy harvest. We evaluated a total of 15 factors in the following domains: biomass, gut transit time (small intestinal and colonic), ileocecal passage pH, SCFAs (circulating and fecal acetate, propionate, butyrate, total, and acetate to propionate ratio), and total GLP-1. We elected not to include relative microbial abundance data given the inability of a general linear model to appropriately handle zero-inflated data. We eliminated independent variables that were highly correlated to each other and selected only variables with reasonable correlations with host metabolizable energy ($P$ value < 0.2 for Pearson or Spearman correlation coefficients) for inclusion in a general linear regression variable selection procedure. For variables from same "family" (e.g., SCFA and microbial species), we selected the variable most highly correlated with host ME. Relevant to our results, fecal acetate and butyrate were highly intercorrelated with propionate. Since propionate had the strongest correlation with host metabolizable energy, it was selected as the representative SCFA. For variables not from the same family (e.g., colonic transit, SCFA), we retained variables unless correlations prohibitively high (i.e., Pearson or Spearman's rho ≥0.75). Based on these criteria, from our original list of 15 variables, three variables (CTT, fecal propionate, and biomass) were included in a stepwise linear regression selection procedure (PROC GLMSELECT in SAS). The selection procedure primarily used $P$ values to determine which variables should be included or excluded from stepwise models, and the model with the lowest Bayesian Information Criterion (BIC)[96] was chosen as the final model.

Statistical analyses were performed using SAS 9.4 and R 4.2.2. A $P$ value less than 0.05 was considered statistically significant. When using the false discovery rate (FDR)[97] to correct for multiple comparisons for differential abundance analysis of gut microbial composition and associations of gut microbes with host metabolizable energy, an FDR $Q$ value < 0.05 was considered statistically significant. Since inadvertent visualization of the diets could unblind investigators, the study could not be conducted in a blinded manner. However, the treatment assignment was not revealed to investigators and all data were collected and maintained blinded to treatment assignment until the database was locked upon study enrollment completion and statistical analyses were completed.

### Reporting summary

Further information on research design is available in the Nature Portfolio Reporting Summary linked to this article.

## Data availability

Source data are provided with this paper. The raw and processed metagenomic sequence data generated in this study have been deposited in the BioProject database under accession code PRJNA913183 PRJNA947193. The following analytical tools (with respective version numbers) were used in this manuscript: ComplexHeatmap 2.14.0; FastQC 0.12.0; GGplot2 3.4.1; HG28 GRCh38.p14; MaAsLin2 1.12.0; MetaPhlAn3 3.0.14; Microbiome 1.20.0; Phyloseq 1.42.0; R 4.2.2; SAS 9.4; TrimGalore 0.6.5; Vegan 2.6-4. Source data are provided with this paper.

## Code availability

The SAS code for the linear mixed model used to analyze all the clinical data is publicly available here: https://zenodo.org/badge/latestdoi/634925145[25]. The code for the multicompartment transit, reaction, and absorption mathematical model used in this manuscript is publicly available here: https://zenodo.org/badge/latestdoi/634925145[25].

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

## Acknowledgements

We thank our study participants, without whom this work would not have been possible. We are grateful for the expert data generation and statistical analyses from the core facilities at the Translational Research Institute. We thank Martin Sadilek, PhD, for his expertise and collaboration in establishing the parameters of the polyethylene glycol assay. Research reported in this publication was supported by the National Institute of Diabetes and Digestive and Kidney Diseases of the National Institutes of Health under Award Number R01DK105829 (S.R.S. and R.K.B.). The content is solely the responsibility of the authors and does not necessarily represent the official views of the National Institutes of Health.

## Author contributions

S.R.S., R.K.-B., K.D.C., and B.E.R. designed the study. K.D.C. and E.A.C. executed the trials and supervised data-generating cores. F.Y., D.I., and B.D. performed statistical analyses. A.M. and T.D. performed mathematical modeling supervised by B.E.R. and R.K.-B. B.D. performed metagenomic analysis, supervised by R.K.-B. K.D.C. wrote the paper with critical revision and input from E.A.C., B.D., D.I., F.Y., A.M., T.L.D., R.E.P., B.E.R., R.K.-B., and S.R.S. S.R.S. provided overall supervision for the clinical trial execution. S.R.S., R.K.-B., and B.E.R. secured grant funding.

## Competing interests

The authors declare no competing interests.

## Additional information

[1]AdventHealth Translational Research Institute, Orlando, FL, USA. [2]Biodesign Center for Health through Microbiomes, Arizona State University, Tempe, AZ, USA. [3]Biodesign Swette Center for Environmental Biotechnology, Arizona State University, Tempe, AZ, USA. [4]Skyology Inc, San Francisco, CA, USA. [5]School of Sustainable Engineering and the Built Environment, Arizona State University, Tempe, AZ, USA. ✉e-mail: Dr.Rosy@asu.edu; Steven.R.Smith@AdventHealth.com

