## [Peer Review File · Nature Communications]

REVIEWER COMMENTS

Reviewer #1 (Remarks to the Author):

The authors present the primary results and additional correlative analyses of a clinical trial evaluating the effect of a dietary intervention which potentially “enhances” the microbiome in comparison to a western diet. The overall design is a cross-over wherein 17 patients are randomized. The primary endpoint is within patient change in fecal energy. The investigators estimate a (significant) difference in change in energy of 116 kcal based on their primary analysis with secondary analyses focused on additional endpoints and the microbiota. The findings are generally interesting and strength is the controlled environment, but there are a number of curious aspects in the protocol which would have raised some flags at my institution (particularly the evaluation of the data at 5 pts which seems highly problematic). In addition, the sample size is modest and there are some analytic issues that require redress. Comments follow.

1. The sample size of the study is quite modest and with only 17 patients, it’s not clear if the properties underlying randomization have even kicked in.
2. There was what appears to be an interim look after 5 patients (even if not formally presented as such). Some further discussion of this would be helpful.
3. It would be helpful to see the baseline characteristics of the patients in each arm (ordering) and before each diet, particularly in terms of microbiome composition.
4. Some additional information on what was happening in the 14 day wash out would be helpful. Is this sufficient for a true washout? This is related to the previous comment and some preliminary evidence would be helpful.
5. The primary endpoint is appropriately analyzed using mixed models. However, some of the microbiome endpoints (e.g. beta diversity analysis, correlatons, etc.) do not respect the broader design and accommodate cluster correlation. These need to be redone using appropriate mixed model-based approaches.

6. maaslin makes strong parametric assumptions regarding the outcome distributions. It would be helpful to verify these findings using something simpler, e.g. a simple LMM on appropriately transformed data (possibly CLR).

7. The use of CAP for ordination and subsequent testing seems curious. My understanding is that CAP is based on LDA which is a supervised approach. Consequently, full permutation of the procedure is necessary for assessing significance. The use of PERMANOVA is thus not appropriate (in addition to the problem with dependent data). The ordination plots are similarly potentially over-optimistic.

8. While I can see that there is a difference between arms in terms of the outcome, it would be helpful to further interpret this clinical effect. How does this magnitude compare with other things?

9. The discussion on missing data is curious as its not clear that the missing data was appropriately accommodated in this case. Complete case analysis does not seem appropriate for dealing with nonignorable missingness. Further, the use of last-observation carried forward has not been regarded as generally appropriate for at least the last 15-20 years. More justification is needed or the use of appropriate analytic models – though the modest sample size may prove somewhat problematic here.

Reviewer #2 (Remarks to the Author):

The paper describes a controlled human intervention trial investigating how a diet (high in dietary fibre and resistant starch, and containing large food particle size and limited quantities of processed food) compared to a Western diet (low in dietary fibre) affects energy balance in humans (energy intake, energy expenditure and energy output). The study is well designed and addresses an important question in the field; namely whether the gut microbiome may contribute to energy balance through energy harvest. I do however have some concerns, which should be addressed.

Main comments

1. It was recently suggested that intestinal transit time is associated with gut microbiome-dependent energy extraction (Boekhorst et al 2022; *Microbiome*, <https://doi.org/10.1186/s40168-022-01418-5>). The authors also find that their mathematical model improves when taking transit time into

account – however, it is unclear how. It would be great if the authors could discuss/explain the role of transit time a bit more? And potentially also compare their results to the results of the study published in *Microbiome*.

2. While colonic transit time was not affected by diet, I was wondering whether the inter-individual differences in host metabolizable energy were related to differences in pH/transit time? How much of the variability was explained by CTT? (The text currently says: “using the CTT explained some of the variability in host metabolizable energy”)

3. The authors measured pH and transit time throughout the GI tract using SmartPills. However, it was unclear to me when these measurements were obtained and whether the authors did any standardization (in terms of timing, meals, etc) before these capsules were swallowed? Furthermore, did you check for correlations between proximal/distal pH and faecal/serum SCFAs? And was the regional pH stable within individuals?

4. The authors describe that the MBD included large food particle size. Could you specify that further - what is large food particle size? Was meal size/calorie intake adjusted to body size or did all participants consume the same amount of energy? Also it is unclear how adherence was assessed. More information (or examples) of what the diets consisted of would be great.

5. The authors evaluated stool consistency using the Bristol stool Scale before the study was initiated. However, was stool consistency measured throughout the study as a proxy to evaluate differences in transit time? Alternatively, the authors may consider measuring the faecal water content as an objective measure to evaluate whether transit time/stool consistency changed. SmartPill transit time may be quite different, due to its large size.

6. The authors report that bacterial biomass increased with the MBD. Given the importance of quantitative microbiome profiling (<https://www.nature.com/articles/nature24460>), the authors should assess diet-induced changes in microbiome abundance using knowledge on total bacterial cell counts as well, and link the absolute microbiome profiles to inter-individual variations in the host metabolizable energy. This could be an important addition.

7. I think it is great that authors managed to measure methane in a whole-room calorimeter. Yet, this part on methane production was still a little unclear to me, since this information was not really incorporated or compared with other obtained data. Typically, only one out of three humans produce methane. Did the authors investigate whether participants differed in colonic energy

extraction depending on methane production? Did the authors study the presence and abundance of methanogens in the microbiome data?

8. I am a little puzzled by the title of the manuscript. I think the study demonstrates that the MBD compared to WD affects the energy harvest/excretion. Yet, what is the evidence that this is mediated by the reprogramming of the human gut microbiome? Some associations to relative abundances are reported, but the causal evidence is not obvious.

9. It would be great if you could discuss the energy that is left in faeces. Is the differences observed between dietary interventions due to differences in energy extraction or differences in bacterial biomass?

10. The authors argue that butyrate-producing bacteria responded to the dietary intervention. Did you observe any correlations between the bacterial group responding to the intervention and changes in SCFAs?

11. Line 132-136: Based on the changes in the gut microbiome, how did you conclude that the microbiota signature that defined the response to the MBD channeled more energy to the microbes (instead of the host) whereas the WD led to conditions in which the gut microbes were "starved"??

Minor comments

- Please specify throughout the manuscript where the hormones were measured. I assume it was measured in blood. Did the measurements in hormones relate to differences in SCFAs?

- It would encourage the authors to upload their sequencing data to public repositories.

- The scale of the colour-gradient showing relative abundance (RA) in figures is a little unclear to me. Maybe you can explain/specify this better (see Fig 2, see Fig 4).

I hope these comments/suggestions will help you improve the manuscript further.

Best wishes,

Henrik Roager

Reviewer #3 (Remarks to the Author):

Thank you for the opportunity to review the paper by Smith et al. This was a highly controlled and rigorous cross-over feeding trial of a Western Diet and Microbiome Enhancer Diet among healthy weight and overweight young to middle-aged adults in the Southeastern U.S. The primary goal was to examine the effect of the diets on the gut microbiome, energy harvest, and host-related factors pertaining to adiposity and enteroendocrine systems. The study demonstrates in an elegant way, host-diet-microbiome interaction. The primary outcome demonstrates that the Microbiome Enhancer Diet is associated with greater fecal energy loss. This finding is relevant to obesity prevention and control. The study contributes significantly to the field given few if any, human dietary intervention trials have been conducted with such rigor to precisely understand diet's effect on the gut microbiome and related host responses. In this Reviewer's opinion, additional information pertaining to the participant's baseline diet pattern and how closely it aligns with the Western or Microbiome Enhancer Diet is relevant. Moreover, sleep-related data is also relevant to characterizing the findings. The methodological details provided including in the attached study protocol allow for reproducibility. In this Reviewer's opinion, the study should be accepted for publication with minor editorial considerations (typos and inconsistent use of capitalization of subheading, abbreviation used but not defined) and with the addition of the participant data detailed above (baseline diet, sleep).

March 22nd, 2023

Manuscript Number: NCOMMS-23-02679-T

Point by Point Response to Reviewer Comments:

We are thankful for the helpful feedback from the reviewers and are pleased that overall, our manuscript was well received by the expert reviewers. We have revised the manuscript in accordance with their recommendations. The manuscript is much improved and better defines the host-diet-microbial interactions that modulate human energy balance.

Our replies to reviewers are presented in blue font, below. In our answers, we indicate the exact text that was changed in the manuscript. Revisions to the manuscript are marked with yellow highlighting. Throughout our responses, below, we provide the full paragraphs for context when appropriate to assist the reviewers in their work.

In addition to the recommended revisions, we

- streamlined the main text in order to adhere to word limits,
- changed the order of reporting to consistently report the microbiome enhancer diet then the western diet,
- edited phrasing to improve clarity,
- now include all Nature required checklist items,
- corrected a few typos and
- made trivial changes to some results due to correct minor rounding errors.

We believe that the revised manuscript will satisfy you and the reviewers and lead to the publication of this important work in your esteemed journal.

Reviewer 1

- 1) The sample size of the study is quite modest and with only 17 patients, it's not clear if the properties underlying randomization have even kicked in.

As discussed in our design paper (<https://doi.org/10.1016/j.conctc.2020.100646> & provided in the review package), the sample size was determined based on the key design features of the study:

- *Crossover design with a dietary washout period so that each individual serves as their own control*
- *Dietary control during the outpatient run-in for both Period A and Period B*
- *Dietary control while in the metabolic ward*
- *Energy balance (based on directly measured energy expenditure in a whole room calorimeter)*
- *Repeated measures of our primary endpoint: 6 days of 24h energy balance observations on each diet*

The statistical power analysis was based upon:

- 1) prior data from our lab on repeated measures (within subject variation) of 24h and sleeping EE (see <https://doi.org/10.1002/oby.23226>), as well as between subject variances;*
- 2) prior measurements of chemical oxygen demand (fecal COD) where replicate variability was known.*

Prima facie, these points, and our power analysis, reveal that our sample size was a priori adequate to detect a 120 kcal/24h difference in EE and an 80 kcal/24-hour difference in fecal energy loss.

We agree that our sample size is not typical of cross-sectional studies that represent the preponderance of human studies of the gut microbiome. Especially important is that we controlled diet and the caloric intake. In essentially all published studies, diet was not provided and energy intake was not controlled. Since diet has a rapid effect on the gut microbiome and under-

overfeeding (number of calories) modifies the gut microbiome, it was essential that we kept the participants in energy balance while giving them a carefully controlled diet.

To restate our argument: While our sample size might not seem adequate for typical microbiome experimental designs, especially observational cross-sectional studies, our sample size was adequate for a cross-over design having strict controls in place: diet composition, energy intake, energy balance, and physical activity. Cross-sectional studies leverage a large sample size to overpower the large experimental errors. Our study's unique features are deep characterization, tight dietary control, and keeping the subjects at energy balance, not a large sample size.

Important to the question about the observed effect of the intervention on energy balance, we performed a retrospective power analysis. That analysis was performed based on the within-subject repeated-measures data derived from the subjects in this experiment. This second power analysis revealed we were powered to detect a 26.5 kcal/24h difference in EE with $n = 17$.

Lastly, regarding the randomization, we randomized by diet sequence in a sex-stratified manner and 8 participants were randomized to the AB sequence vs. 9 participants to BA. Thus, each individual served as their own control. The population was balanced by sex and by design, individuals were generally healthy, without obesity and they were similar in age.

For *all* comparisons, we also tested for the effects of order (i.e., diet sequence) and period. We did not find any statistically significant effects of sequence or period, supporting no appreciable carryover effect.

We have added these details to lines 653-659 as follows:

Statistical analyses. Details of the sample size determination have been previously published¹². Briefly, our prior modeling experiments predicted a difference of 110 kcal of fecal energy loss (COD g/day) on the MBD vs. WD for a person consuming 2,000 kcal/day. Our power analysis was based on prior data on repeated measures of 24-hour and sleeping energy expenditure⁸⁸ and prior COD measurements where replicate variability was known. Our sample size was adequate to detect a 120 kcal/day difference in energy expenditure and an 80 kcal/day difference in fecal COD. This small sample size was made possible by the repeated measures, precision of our measurements, crossover design, and tight control of diet and environment.

- 2) There was what appears to be an interim look after 5 patients (even if not formally presented as such). Some further discussion of this would be helpful.

We had originally intended to measure chemical oxygen demand (fecal energy content) and polyethylene glycol (internal 'spike') from the first 5 participants. The purpose of looking at the first 5 was to ensure our initial power estimates were correct by assessing replicate variability. This quality-control procedure was included in the original version of the research protocol. However, we **did not** perform this QC check and **did not** look at any unblinded data before locking the dataset and executing the statistical analysis. Principal secondary clinical outcome variables were prioritized and analyzed sequentially – also in a blinded fashion - using a single statistical model and SAS code.

The manuscript has been updated to reflect that all data were blinded until the database was locked (lines 700-701).

- 3) It would be helpful to see the baseline characteristics of the patients in each arm (ordering) and before each diet, particularly in terms of microbiome composition.

Baseline characteristics of the study population are presented in **Supplementary Table 4**.

As discussed in Johnson et al. (<https://doi.org/10.3389/fnut.2020.00079>), baseline (free-living) samples are useful in certain study designs, such as longitudinal observational studies where methodologies are insufficient (such as from lack of dietary and other environmental controls) to identify inter-individual variation or when, a priori, the analysis is intended to be pre- and post-diet intervention in a parallel arm design. For domiciled metabolic ward crossover studies such as ours, Johnson et al. **do not** recommend a free-living sample. They state that interventional crossovers are “the most optimal study design for studying the effects of a dietary intervention because each participant acts as their own control”.

Adding to their comments, we respectfully submit that a free-living sample without antecedent dietary or environmental controls could only be correlated to the self-reported dietary intake data. Self-reported dietary data are widely recognized as

inaccurate and only useful for large population studies where hundreds or thousands of participants will overpower the inaccuracy of self-reported dietary data.

- 4) Some additional information on what was happening in the 14 day wash out would be helpful.
These details of the conditions during the washout were added to the methods section in lines 424-428.

During the 14-day washout, participants returned home and were instructed to resume their usual diet and physical activity. The following additional restrictions were implemented prior to each study period: avoidance of alcohol during outpatient feeding periods, no caffeine up to 72 h before admission, no artificial sweeteners, and no strenuous physical activity 48 h prior to admission.

- 5) Is this sufficient for a true washout? This is related to the previous comment and some preliminary evidence would be helpful.

A 14-day washout is sufficient based on our design and on the literature.

Firstly, with respect to our design, see comment above (item #1) about the lack of order effects. Furthermore, in addition to the 14-day washout, participants consumed their assigned diet for an additional 14 days (11 outpatient and 3 inpatient) before measurement of study endpoints.

Secondly, the literature is clear on how rapidly the gut microbiome responds to diet interventions. In this seminal publication <https://doi.org/10.1038/nature12820> the authors found that the gut microbiome rapidly adjusted to diet shifts (5 days) and reverted to its original structure within 2 days. Thirdly, this review <https://doi.org/10.3390/nu11122862> shows that washout periods in crossover studies ranging from 14 to 28 days are commonly used for these types of studies.

- 6) The primary endpoint is appropriately analyzed using mixed models. However, some of the microbiome endpoints (e.g. beta diversity analysis, correlations, etc.) do not respect the broader design and accommodate cluster correlation. These need to be redone using appropriate mixed model-based approaches.

We were not clear enough in our Methods section about how we analyzed the microbiome endpoints. With one exception, we analyzed all microbiome endpoints using the same linear mixed model (LMM) as we used for the clinical endpoints. The exception was the correlation analysis between host metabolizable energy and microbial abundance, which evaluated the subset of samples from only the MBD using a linear model instead of LMM.

To clarify the application of LMM to microbiome endpoints, we added the following text highlighted in yellow (Lines 577-601):

Species Alpha- and Beta-Diversity. All calculations and analyses were conducted in R (Version 4.2.2)⁷⁶. Taxonomic composition output from MetaPhlan3 was processed for beta-diversity analysis using the “phyloseq” R package (Version 1.42.0)⁷⁷. A rarefaction curve was created using the “vegan” R package (Version 2.6-40)⁷⁸ to determine the optimal count-depth for rarefaction. Once the optimal count-depth was determined, rarefaction was performed using phyloseq. Alpha-diversity metrics were calculated using the “microbiome” R package (Version 1.20.0)⁷⁹. After samples were rarified, each sample had 3,578,445 sequences. Bray-Curtis and Jaccard distance matrices were calculated on the rarefied count data using vegan. The distance matrices were tested for significance by PERMANOVA using vegan. Diet was the only significant term for both metrics (Bray-Curtis: $P = 0.017$, Jaccard: $P = 0.016$). Beta-dispersion was calculated, and the results tested for significance with the ANOVA-like permutation test and Tukey’s HSD in vegan. Constrained Analysis of Principal Coordinates (CAP) ordination was performed with vegan. CAP is a multivariate linear method⁸⁰ that we used to assess how much of the variation in the beta-diversity could be explained by diet. Consistent with the linear mixed model approach used for analyzing the clinical data, diet, sequence, and period were fixed, and participant was a random factor. Statistical significance testing was performed with PERMANOVA in base R. Beta-diversity ordination figures were created using the “ggplot2” R package (Version 3.4.1)⁸¹. Differential abundance heatmap figures were created using the “ComplexHeatmap” R package (Version 2.14.0)⁸².

Differential Abundance. Differential abundance testing by diet and associations of species relative abundance with host metabolizable energy were carried out using the output of MetaPhlan3 in the “MaAsLin2” R package (Version

1.12.0)²⁴. Taxonomic counts were filtered with a 25% prevalence cut-off. Compound Poisson multivariate linear models were used to account for zero-inflated data²⁴. For differential abundance testing, diet, sequence, and period were fixed, with participant as the random factor. For analyzing associations of species relative abundance with host metabolizable energy, metabolizable energy was the fixed independent factor, because the analysis was restricted to samples in the MBD only

- 7) *maaslin* makes strong parametric assumptions regarding the outcome distributions. It would be helpful to verify these findings using something simpler, e.g. a simple LMM on appropriately transformed data (possibly CLR). *Maaslin2* offers a range of different models to accommodate different outcome distributions. We chose the Compound-Poisson Linear Mixed Model (CPLMM), a special type of LMM used for data with many entries that have small or zero values. Traditional LMM does not generate reliable results with sparse data. For this reason, the CPLMM is particularly well-suited to handle our sparse microbiome abundance data, which contain ~75% zeros (as expected with relative abundance data). Given the high presence of data with zeros and the biological relevance of the taxa identified in our CPLMM analysis, we are confident that we employed the best approach for analyzing our data.
- 8) The use of CAP for ordination and subsequent testing seems curious. My understanding is that CAP is based on LDA which is a supervised approach. Consequently, full permutation of the procedure is necessary for assessing significance. The use of PERMANOVA is thus not appropriate (in addition to the problem with dependent data). The ordination plots are similarly potentially over-optimistic.

CAP (Constrained Analysis of Principal Coordinates) is not based on LDA. Instead, CAP is a distance-based variant of RDA (Redundancy Analysis) and is based on PCoA (Principal Coordinates Analysis). CAP is also known as distance-based RDA (dbRDA). Although CAP and LDA are supervised linear models, they have different goals. On the one hand, LDA is used to classify response variables into different groups, and it can be used for prediction. CAP, on the other hand, is used to explain the variation of response variables using a set of explanatory variables; CAP also can account for random effects. Thus, CAP allowed us to use an LMM to determine if beta-diversity differed by diet.

Because CAP is not a classification method, it only requires significance testing of the association of the response variables with the explanatory variables. In this case, our use of PERMANOVA is appropriate, because PERMANOVA is a distance-based test for the association of a beta-diversity metric with explanatory variables. In the initial steps of our beta-diversity analysis we used PERMANOVA to test for significance directly on the Bray-Curtis and Jaccard distance matrices and found that diet was the only significant term for both metrics.

The ordination plots are accurate for CAP analysis. CAP constrains its explanation of response-variable variation to the explanatory variables included in the analysis: i.e., diet, sequence, and period in our study. CAP ordination plots, therefore, only show response-variable variation due to the explanatory variables we tested.

We are convinced our analysis is the right one, but we did add a sentence in the methods section giving more detail about the results of the initial PERMANOVA testing on the distance matrices. Added text is highlighted in yellow (Lines 584-585):

*The distance matrices were tested for significance by PERMANOVA using *vegan*. Diet was the only significant term for both metrics (Bray-Curtis: P = 0.017, Jaccard: P = 0.016).*

- 9) While I can see that there is a difference between arms in terms of the outcome, it would be helpful to further interpret this clinical effect. How does this magnitude compare with other things?

This is an important question. In this paper, <https://doi.org/10.1002/oby.20813>, Gobel et al. developed a model that quantified metabolizable energy intake in response to obesity pharmacotherapy (weight loss). When weight reaches its nadir, metabolizable energy intake reductions can be as large as ~1,500 kcal/day. When counter-regulatory mechanisms affecting energy intake are activated, energy intake increases, and some weight is regained. At that time, metabolizable energy intake necessary to maintain a reduced weight is ~200 kcal/day. Therefore, the additional 116 kcals/day lost in feces in our study is clinically meaningful.

In addition, there is an entire literature (e.g. Hill et al <https://doi.org/10.1159/000345030>) on the population level impact of “small changes”.

We have now more clearly described in the discussion section (lines 300-305) as follows:

The greater fecal energy loss translated to an additional 116 kcal/day lost in feces when participants were fed the MBD as compared to the WD. The clinical significance of this difference can be inferred from the reduction in food intake needed to maintain a weight-reduced state with obesity pharmacotherapy. The reduced energy intake (negative energy balance) needed to maintain this new weight-reduced state has been modeled to be approximately 200 kcal/day⁴⁴. In addition, the cumulative impact over time of a 116 kcal/day energy deficit is in alignment with the population-level impact of “small changes” in energy balance to promote weight loss⁴⁵.

- 10) The discussion on missing data is curious as its not clear that the missing data was appropriately accommodated in this case. Complete case analysis does not seem appropriate for dealing with nonignorable missingness. Further, the use of last-observation carried forward has not been regarded as generally appropriate for at least the last 15-20 years. More justification is needed or the use of appropriate analytic models – though the modest sample size may prove somewhat problematic here.

We clarify that we used an interpolation method to impute the single missing values for 3 participants in the enteroendocrine hormone data. We did not carry forward the last observation for enteroendocrine hormones.

The reviewer's concerns about LOCF are valid as it can introduce bias if the underlying assumptions are not scientifically sound (<https://doi.org/10.17226/12955>), particularly when modeling imputation of 30% of data points.

We used LOCF only for ONE (1) timepoint (+180) from the acetaminophen (i.e., gastric emptying) test for ONE (1) participant who did not complete the final timepoint (data missing not at random).

This represents an imputation of 1 out of 204 timepoints (0.5%). The LOCF method was appropriate in this case because of the relative stability of the +120 min and +180 min measurements across the remaining participants (median difference 0.55 ng/mL; IQR -0.55, 1.55).

Lastly, as there was no difference in gastric emptying by diet, these data are only presented in the supplementary data and distant from the main thrust of the paper. Therefore, our assumptions are scientifically sound, making this use of LOCF for a single timepoint valid. These details were added to lines 665-677 as follows:

Appropriate to our randomized crossover design, we used a linear mixed model (SAS PROC MIXED) with diet, period, and sequence as fixed effects and participant as a random effect to compare differences by diet in our primary endpoint (host metabolizable energy: fecal energy loss adjusted to energy intake) and most other secondary and exploratory endpoints. When the distribution of the model residuals was found to deviate considerably from normality, a logarithmic transformation was applied. For each endpoint, we included only participants with complete data for both diet interventions when the data were considered to be missing not at random. Three values that were considered to be missing at random for the enteroendocrine hormone data (one out of 18 serial timepoints over the day was missing for each of three participants due to temporary issues with blood draw or laboratory analysis, but not because the entire sample was missing) were imputed using the interpolation method (i.e., averaging the previous and subsequent values)⁸⁹. We imputed the last (+180) timepoint of 6 serial timepoints for one participant for the gastric emptying assay using the last observation carried forward approach, given the relative stability of the +120 min and +180 min timepoints (median difference 0.55 ng/mL; IQR -0.55, 1.55)⁸⁹.

Reviewer 2

The paper describes a controlled human intervention trial investigating how a diet (high in dietary fibre and resistant starch, and containing large food particle size and limited quantities of processed food) compared to a Western diet (low in dietary fibre) affects energy balance in humans (energy intake, energy expenditure and energy output). The study is well designed and addresses an important question in the field; namely whether the gut microbiome may contribute to energy balance through energy harvest. I do however have some concerns, which should be addressed.

Main comments

- 1) It was recently suggested that intestinal transit time is associated with gut microbiome-dependent energy extraction (Boekhorst et al 2022; Microbiome, <https://doi.org/10.1186/s40168-022-01418-5>). The authors also find that their mathematical model improves when taking transit time into account – however, it is unclear how. It would be great if the authors good discuss/explain the role a transit time a bit more?

Your question led us to realize that we overinterpreted the utility of our *in silico* model to explain the role of CTT. In brief, we used our established model (Marcus et al., 2021) and the measured colonic transit times (CTT) to calculate the first-order overall hydrolysis rates within the colon when assuming hydrolysis is the rate-limiting step for the fermentation of protein and carbohydrates. In the original version of the model, the CTT was assigned a constant value of 48 h, a typical value. Our study results showed the CTT for participants on both diets had a large range of CTT times, from 4 to 93 hours. Thus, a constant 48 hours is probably an inaccurate assumption of hydrolysis rates.

Incorporating measured colonic transit time reduced the bias in our model. Upon further review, we identified an error in the calculation of the concordance correlation coefficient (CCC) and now provide the data for the components of concordance correlation. These data, in conjunction with the Bland-Altman plots, clearly show the reduction in bias.

We revised our figures as shown below:

Fig. 4b-c

Fig. 4. The contributions of the gut microbiome to host metabolizable energy. . b, Concordance correlation coefficient plot between predicted (by modeling) and measured host metabolizable energy (ME) using the same fixed CTT (48 h) for all participants. Dashed line is simple linear regression between pairs of data; solid line is identity line (perfect reproducibility actual and modeled data). c, The same plot with each participant's measured CTT.

To clarify the role of CTT in the model, we added the following text highlighted in yellow:

Results (Lines 254-261)

Our previously published model used a fixed CTT of 48 hours, which is a reasonable population-level estimate for healthy adults⁴². With a fixed CTT, the mean modeled host metabolizable energy for participants on the MBD was $92.4 \pm 0.001\%$ and for WD was $95.2 \pm 0.001\%$ (Fig. 4b). This is similar to the mean host metabolizable energy we measured on the MBD and the WD ($89.5 \pm 0.73\%$ and $95.4 \pm 0.21\%$, respectively; Fig. 1b). However, the model was biased as evidenced by the linear distribution of the points which estimated essentially the same metabolizable energy for each person in contrast to the variability in the measured metabolizable energy (Fig. 4b) and Bland-Altman plot (Supplementary Fig. 5c). We hypothesized that we could reduce the model's bias by incorporating measured CTT since it is a key modulator of microbial composition, fermentation, and host energy balance³³. Incorporating the measured CTT values reduced bias based on greater reproducibility (concordance correlation coefficient: 0.645 with fixed CTT and 0.983 with measured CTT) and accuracy (bias correction factor: 0.514 with fixed CTT and 0.789 with measured CTT; Fig. 4b-c). Furthermore, systematic and proportional biases were minimized, as shown in the Bland-Altman plots (Supplementary Fig. 5d). Collectively, these data suggest that CTT is an important factor for estimating host metabolizable energy.

Methods (Lines 644-649)

We then improved the model by using the measured CTT and evaluated the impact of this change by comparing actual versus modeled data. The performance of the model for estimating host metabolizable energy was evaluated by analyzing concordance correlation coefficient components: bias correction factor (Cb)- accuracy; Pearson's correlation coefficient (r) between measured and modeled metabolizable energy- precision; concordance coefficient correlation (ρ_c)- precision⁸⁶. We evaluated systematic and proportional bias with a Bland-Altman plot⁸⁷.

Importantly- we focused and clarified the model results on the two key findings:

1. predicted energy in fecal biomass accounts for >25% of the total fecal energy (on both diets). This is discussed below (Item #15)
2. On the MBD there is a modeled 2X greater absorption of SCFAs, even though total energy absorbed is lower. The SCFA absorption is well defined in the original manuscript.

- 2) And potentially also compare their results to the results of the study published in Microbiome.

Boekhorst et al. found a positive correlation between a higher colonic transit time (CTT) and a lower energy harvest efficiency, which was the opposite of their original hypothesis. We cannot quantitatively compare results between the two studies, which used different study designs, data stratifications, and units of measurements.

Most importantly, we used measured fecal COD normalized to the total fecal weight for all 6 days AND corrected for a PEG internal standard.

Boekhorst et al. used the measured fecal energy normalized to the dry fecal weight without any internal standard to control for differences in collection time of the sample. Without an internal standard for normalization, the analysis of the results does not contextualize the fecal energy in terms of the dietary intake. Thus, a direct comparison between our study results and Boekhorst et al. isn't possible.

We briefly described the strengths of our approach on as follows:

Results lines 96-102

The overall goal of our study was to modulate the gut microbiome and employ a quantitative paradigm with enough precision to detect within-participant responses to the diet intervention. A key contribution to understanding the role of the gut microbiome on energy balance involves fecal energy. Prior methods lack precision and often provide results as energy per gram of feces, which does not contextualize fecal energy in terms of dietary intake (which we precisely controlled) and makes it difficult to interpret the relationships of fecal energy to host phenotypes²⁰. To this end, according to the method of Pak²¹, we administered a low, non-laxative dose of non-absorbable non-digestible polyethylene glycol (PEG) with each meal.

Results lines 176-187

The relationships among diet composition, gut microbes, and colonic transit time (CTT) are complex, multi-directional, and vary within individuals over time and between individuals³³. Given the potential importance of CTT on the microbiota-driven host response to dietary manipulations, we evaluated whole-gut transit using a pH-sensing radiotransmitter device. This device has advantages to other methods (such as the use of scintigraphy or radio-opaque markers) including that it is noninvasive, generates pH, temperature and pressure data, provides whole gut and regional data, and importantly, the test is standardized to improve reliability of interindividual and longitudinal assessments³⁴. In addition, the assessment was done under conditions of energy balance and with controlled diets that were customized to meet exactly the needs of each participant. This differs from other approaches that have evaluated gut microbiome-CTT interactions³⁵ and is an important advancement given the critical role of diet composition and quantity on both the gut microbiome and CTT. We did not find a statistically significant difference in CTT by diet (29.7 ± 4.4 hours on MBD vs. 39.2 ± 6.2 hours on WD; $P = 0.14$; Fig. 3e).

- 3) While colonic transit time was not affected by diet, I was wondering whether the inter-individual differences in host metabolizable energy were related to differences in pH/transit time? How much of the variability was explained by CTT?

(The text currently says: “using the CTT explained some of the variability in host metabolizable energy”)

To address these points, we implemented a new approach. To gain a better understanding of host and microbial factors that may explain part of the variability in host metabolizable energy (ME) (including pH and SCFAs) on the microbiome enhancer diet (MBD), we undertook a model selection procedure. In preliminary steps, we first identified measured variables with known biological links to host ME and correlated them with host ME on the MBD. Variables that were correlated with host ME ($p < 0.2$ for Spearman or Pearson's rho) were considered in a second correlation matrix. Based on the second correlation matrix, we eliminated independent predictor variables of host ME that were highly correlated to each other. For variables from same “family” (e.g., SCFA), we selected the variable most highly correlated with host ME. For variables not from the same family (e.g. colonic transit, SCFA), we retained variables unless correlations prohibitively high (i.e., Pearson or Spearman's rho ≥ 0.75).

Based on these preliminary steps, 3 variables moved forward to a linear regression model: colonic transit time, fecal propionate and bacterial copy number (a surrogate for biomass). The model selection procedure (PROC GLMSELECT in SAS) used p-values and Bayesian Information Criterion (<https://doi.org/10.1371/journal.pone.0206711>) to specify a final model with up to three variables, given our sample size. The selection procedure selected fecal propionate and biomass into the final model and revealed that these two variables jointly explain 58% of the variance in host metabolizable energy.

We describe this in the manuscript as follows:

Results on lines 272-284

The wide interindividual variability in host metabolizable energy in response to the MBD is likely related to a combination of host and microbial factors. We postulated that we would be able to identify key parameters within our study sample that explain a portion of this variability. Thus, we undertook an exploratory, multi-step statistical process to identify potential host and microbial determinants of host metabolizable energy using data from the MBD only. We did so with an aim to consider a limited number of determinants of host metabolizable energy, given our sample size. Following consideration of 15 potential determinants as described in Methods, the final step in this process was a multivariate selection procedure into which one host factor (CTT) and two microbial factors (fecal propionate and biomass) were entered. The selection procedure chose fecal propionate and biomass—both microbial factors—for the final optimized model and revealed that these two variables jointly explain 58% of the variance in host metabolizable energy. According to the final model, each SD increase in 6-day fecal propionate (858.6 mg) was associated with 2.1% less host metabolizable energy (95%CI 0.95, 3.2), while each standard deviation increase in biomass (0.49 log of 16S gene copy number) was associated with 1.6% less host metabolizable energy (95%CI 0.44, 2.7) (Fig. 4f).

Fig 4f

Fig. 4. The contributions of the gut microbiome to host energy harvest. f, Scatterplot of predicted and measured host ME on the MBD; predicted host ME was obtained from the model selection procedure which estimated that 6-

day fecal propionate and biomass jointly explained 58% of the variance in host ME. Thus, the R-squared for the simple linear regression of predicted and measured host ME is 0.58.

Discussion on lines 368-370

In addition, we implemented a statistical approach and found that microbial biomass and fecal propionate (as a representative of colonic fermentation) explained over half of the variance in host metabolizable energy.

Methods on lines 679-695

To identify host and microbial determinants of host metabolizable energy, we first constructed a correlation matrix with hypothesis-driven host and microbial factors that might influence the efficiency of dietary energy harvest. We evaluated a total of 15 factors in the following domains: biomass, gut transit time (small intestinal and colonic), ileocecal passage pH, SCFAs (circulating and fecal acetate, propionate, butyrate, total, and acetate to propionate ratio), and total GLP-1. We elected not to include relative microbial abundance data given the inability of a general linear model to appropriately handle zero-inflated data. We eliminated independent variables that were highly correlated to each other and selected only variables with reasonable correlations with host metabolizable energy (P value < 0.2 for Pearson or Spearman correlation coefficients) for inclusion in a general linear regression variable selection procedure. For variables from same “family” (e.g., SCFA and microbial species), we selected the variable most highly correlated with host ME. Relevant to our results, fecal acetate and butyrate were highly intercorrelated with propionate. Since propionate had the strongest correlation with host metabolizable energy, it was selected as the representative SCFA. For variables not from the same family (e.g., colonic transit, SCFA), we retained variables unless correlations prohibitively high (i.e., Pearson or Spearman’s rho ≥ 0.75). Based on these criteria, from our original list of 15 variables, three variables were included in a stepwise linear regression selection procedure (PROC GLMSELECT in SAS). The selection procedure primarily used p-values to determine which variables should be included or excluded from stepwise models, and the model with the lowest Bayesian Information Criterion (BIC)⁹⁰ was chosen as the final model.

- 4) The authors measured pH and transit time throughout the GI tract using SmartPills. However, it was unclear to me when these measurements were obtained and whether the authors did any standardization (in terms of timing, meals, etc) before these capsules were swallowed?

The SmartPill™ was administered while participants were in the whole-room calorimeter under a standardized protocol. It was consumed right after breakfast.

The details above are now added to the methods section starting on lines 617-618.

Participants consumed the assigned diet for at least 15 days prior to administration of the SmartPills. Furthermore, while in the calorimeter, all meals were given at the exact same time of day.

*A calorimetry schedule of events was added as **Supplementary Table 2** to assist the reader.*

- 5) Furthermore, did you check for correlations between proximal/distal pH and faecal/serum SCFAs? And was the regional pH stable within individuals?

We have another manuscript in preparation that further explores associations amongst gut transit, pH, and SCFAs. That is a comprehensive analysis that is far beyond what we can include in this manuscript focused on our topline results.

- 6) The authors describe that the MBD included large food particle size. Could you specify that further - what is large food particle sizes?

*Details of all diet drivers can be found in the publication where we describe the design in detail:
<https://doi.org/10.1016/j.conctc.2020.100646>.*

For example, we selected foods such as whole nuts on the MBD and nut butters on the WD. The rationale behind this is that fine grinding of foods increases energy availability to the host in the small intestine.

See also <https://doi.org/10.1093/nutrit/nux072>

Large particles allow more of the ingested food to reach the large intestine making energy more available for the microbes, which are in the large intestine <https://doi.org/10.1016/j.jff.2017.12.035>.

Brief details on added to the paper (lines 458-466) as follows:

Dietary intervention. Our study diets were designed to maximize the differences of dietary substrate availability to gut microbes with the MBD while minimizing it with the WD. To achieve this, the MBD was higher in fiber and resistant starch, which are known substrates for microbial fermentation¹⁴. We also provided larger food particles (whole nuts vs. nut butter, for example) because fine grinding of foods makes nutrients more bioavailable to the human host and thus, less available to the gut microbes^{66,67}. One final element of our diets was minimizing processed foods on the MBD, in contrast to the known excess of processed foods in the WD. Accumulating evidence indicates that processed foods, in addition to lacking fiber and having smaller particle sizes, negatively impact host health in part via the gut microbiome⁶⁸. Details of the diet design, including sample menus can be found in our trial design publication¹².

- 7) Was meal size/calorie intake adjusted to body size or did all participants consume the same amount of energy?

Thank you for this important question, as this is a key element of our design that has a major bearing on the interpretation of the results.

All participants received exactly the dietary energy needed to maintain energy balance (energy in = energy out). In other words- the diets were customized not just to body size, but to energy expenditure, which is driven primarily by lean mass.

For the outpatient feeding period, we used armband accelerometry to determine energy needs. We monitored weight throughout the study as a proxy for energy balance.

For the metabolic ward studies, we measured energy expenditure with a whole room calorimeter once prior to the start of each study period. These data were used to define energy intake for the first of 6-days in the calorimeter when the primary endpoint was measured. Provided energy was then adjusted as needed on a daily basis so that each day each participant was at energy balance.

The goal for these 6 days (which was achieved- see **Supplementary Fig. 2a**) was to keep participants in energy balance during the measurement of the primary endpoint. The details of the methodology for achieving net-zero energy balance while in the whole room calorimeter are described in this publication: <https://doi.org/10.1016/j.conctc.2020.100646>.

Our implementation of an energy balance experimental paradigm is in contrast to essentially all prior publications on gut microbiome-diet interactions which did not control for dietary energy intake or precisely measure energy expenditure to ensure energy balance was achieved equivalently with all interventions.

Thus, our study sets a new standard for the field by controlling dietary intake (energy and composition) and quantifying food and fecal energy per unit time. Prior studies have large errors of measurement and interpretation because fecal energy was measured per gram of fecal matter which is variable depending on moisture content, and does not provide appropriate time information to link with intake.

These details were included original manuscript, but we added clarifying phrases. One such phrase is exemplified below and highlighted in yellow (line 63).

To avoid the confounding effects of energy imbalance on host and microbial metabolism, the diet intervention maintained each participant in energy balance. Energy balance, evaluated by real-time energy intake (**personalized to the energy needs of each participant¹²**) and energy expenditure (measured via whole-room indirect calorimetry), was maintained within our target of +/- 50 kcals per 6-day calorimeter stay (WD 4.1 ± 5.1 kcal/day; MBD 5.4 ± 2.8 kcal/day; $p = 0.8$; Supplementary Fig. 2a). Weight stability was a secondary criterion for evaluating energy balance,

and we previously reported that weight was stable during the 6-day calorimetry assessment period whilst the primary endpoint was measured **and data was being generated** blinded to the diet assignment¹².

- 8) Also it is unclear how adherence was assessed.

Adherence procedures throughout various parts of our study are described in the published clinical trial design paper <https://doi.org/10.1016/j.conctc.2020.100646>. Briefly, all foods were prepared in our metabolic kitchen and provided to study participants for the duration of the study and 100% consumption was required. During the outpatient controlled feeding, participants came to our study site twice per week to pick up meals, bring back uneaten food (if any), and to consume one meal while in-house. During the inpatient phase, meals were monitored to ensure adherence. Uneaten food during both phases (if any) was weighed back and actual intake was calculated.

These details were included in the original manuscript as follows (current lines 468-480):

Diets were prepared in our metabolic kitchen based on kcals needed to maintain energy balance as determined by whole-room indirect calorimetry. Diets were designed with menu software (ProNutra Version 3.5, Viocare, Inc, Princeton, NJ) that proportionately calculated diets based on **each participant's** energy needs. Duplicate meals were prepared during all calorimetry days and evaluated for energy content as a quality-control step (Eurofins, Madison, WI). Nutritional composition of the diets was based on the menu software database (USDA Database Standard Reference 23), with the exception of resistant starch, because it is absent from all currently available nutritional databases. We limited foods containing resistant starch on the WD and then estimated the content on both diets based on published estimates of resistant starch content of common foods⁶⁹. Diets were equivalent in metabolizable energy and proportions of macronutrients. As much as possible, we used similar types of foods on both diets to minimize differences in micronutrients. Supplementary Table 2 shows the energy, macronutrient, and drivers in each diet. Consumption of 100% of provided foods was required. Diet adherence was monitored during the 11-day outpatient phase at clinic visits 2 or 3 times per week where at least one meal was consumed on site. During the domiciled metabolic ward phases, all meals were monitored.

This additional detail was added on lines 481-483:

Adherence was calculated by weighing back uneaten food (if any) and recalculating dietary intake as a percentage of provided diet vs. unconsumed diet¹².

*The evidence of adherence is provided in **Supplementary Fig. 2b**.*

- 9) More information (or examples) of what the diets consisted of would be great.

Sample menus are presented in the supplementary appendix of the published clinical trial design paper <https://doi.org/10.1016/j.conctc.2020.100646>. The availability of sample menus was added to lines 482-483.

- 10) The authors evaluated stool consistency using the Bristol stool Scale before the study was initiated. However, was stool consistency measured throughout the study as a proxy to evaluate differences in transit time? Alternatively, the authors may consider measuring the faecal water content as an objective measure to evaluate whether transit time/stool consistency changed.

*The Bristol Stool Scale was measured only at the beginning of the study during the screening period. In future studies, we will measure it longitudinally, as we agree that these data would have been useful to evaluate changes in fecal properties with diet. We point to our adverse events reporting, which did not find frequent gastrointestinal symptoms with either diet (see **Supplementary Table 1**). Measuring whole-gut and segmental transit time with the radiotracer pill, we noted highly interindividual changes between individuals and across diet, which is in alignment with the variability noted in other studies (<http://dx.doi.org/10.1136/gutjnl-2022-328166>). These data are the strongest indicator of changes in transit time.*

Note: regarding the measurement of water: given that we used an internal “spike” [PEG] variation in water content would not affect the main results of the experiments; namely, fecal energy and net metabolizable energy [ME].

- 11) The authors report that bacterial biomass increased with the MBD. Given the importance of quantitative microbiome profiling (<https://www.nature.com/articles/nature24460>), the authors should assess diet-induced changes in microbiome abundance using knowledge on total bacterial cell counts as well, and link the absolute microbiome profiles to inter-individual variations in the host metabolizable energy. This could be an important addition.

*We agree that total bacterial cell counts, if possible, would be an advancement to our manuscript and the field. However, measuring this with the accuracy needed is not an easy task due to the complexity of the method and an inherent limitation: both parallel amplicon sequencing and flow cytometric enumeration of microbial cells (as reported in Vandeputte et. al) and estimation of biomass by qPCR rely on the number of 16S rRNA gene copies per genome in each microbe identified. There is a large variation in 16s rRNA gene copy numbers per genome (from 1 to 21) (<https://doi.org/10.1093/nar/gku1201>). Importantly, 171 of the 277 species identified in this study do not have a reported 16S rRNA gene copy number per genome in the NCBI. To convert our abundance data based on 16S-rRNA-gene qPCR (or cell counts) to absolute abundance, we would need to know the 16S rRNA-gene copy number per genome for all the species detected. As mentioned above, more than half of those species do not have reported 16S rRNA gene copy numbers per genome (<https://doi.org/10.1093/nar/gku1201>). Hence, these species would be left out of the analysis of impacts on host metabolizable energy. In particular, among the species identified with a significant correlation to host metabolizable energy (Extended Data Fig. 5a-b), 5 of those species (*Catenibacterium mitsuokai*, *Dorea formicigenerans*, *Roseburia inulinivorans*, *Clostridium* sp. CAG 58, and *Clostridium boltea*) have no 16S rRNA gene copy-number per genome information and would be excluded. Running an analysis with the large number of missing species surely would lead to inaccurate trends and incorrect conclusions. We avoided this pitfall.*

Although not an exact measure of cell counts or biomass, the 16S rRNA gene copy numbers give a reasonable estimate of trends in microbial biomass, which is how we use these data (in lines 119-122):

Given our primary finding that diet produced a significant change in host metabolizable energy, we next evaluated the microbial phenotype associated with host energy balance. Mean daily fecal weight was higher on the MBD ($P < 0.0001$; Supplementary Fig. 3a), and a proportion of this additional weight was due to a significant increase in 16S rRNA genes ($P < 0.0001$; Fig. 2a), an indication of fecal bacterial biomass increase...

We made no changes to this text.

We include the lack of quantitative bacterial counts as a limitation in the discussion on lines 386-399 as part of a more comprehensive assessment of limitations:

Our study had several limitations that should be considered when interpreting our results. Although our data revealed key gut microbiome contributions to human energy balance, we were unable to deconvolute the complex human host-diet-gut microbiome interactions and therefore, cannot establish whether the changes in energy balance we observed are causally attributable to the diet, the microbes, or some combination. Future directions to address these gaps include implementing bioinformatic pipelines that allow for absolute quantification of microbial species⁶⁰ and the proportion of fecal energy contributed by the gut microbes (vs. undigested food). Additional mechanistic experiments are needed in preclinical models and bioreactors to establish the specific physiological pathways driven by communities of microbes⁶¹, identify systemic lipids, metabolites or proteins that mediate diet-host-microbe interactions, and understand how microbes utilize dietary components⁶². Given the small sample size of our precisely controlled study, selection bias may limit generalizability. Future studies should confirm and expand our findings in larger study samples. Nonetheless, the randomized crossover design vastly reduces the likelihood of confounder bias and yields outstanding internal validity. Larger studies could enable subgroup analyses to inform whether effect sizes vary by sex or other participant characteristics.

- 12) I think it is a great that authors managed to measure methane in a whole-room calorimeter. Yet, this part on methane production was still a little unclear to me, since this information was not really incorporated or compared with other obtained data. Typically, only one out of three humans produce methane. Did the authors investigate whether participants differed in colonic energy extraction depending on methane production? Did the authors study the presence and

abundance of methanogens in the microbiome data?

We appreciate your enthusiasm on the novel methodology of measuring methane over a 24-hour period within a whole room calorimeter. We have asked many of the same questions that you have. Based on your feedback, we concluded that including additional data and figures related to methane, which do not directly address the question of the host-diet-microbiome interactions that modulate host energy balance, would take the manuscript away from the main focus. Therefore, we removed methane from this paper, and we will write a full manuscript on the novel dataset (now in preparation).

- 13) I am a little puzzled by the title of the manuscript. I think the study demonstrates that the MBD compared to WD affects the energy harvest/excretion. Yet, what is the evidence that this is mediated by the reprogramming of the human gut microbiome? Some associations to relative abundances are reported, but the causal evidence is not obvious.

We revised the title as follows: “Human Host-Diet-Gut Microbiome Interactions Influence Energy Balance”. This now reflects the complex and integrative nature of our observed results that were within a paradigm in which the same host was exposed to two controlled diets. This drastically changed the gut microbiome (biomass, beta diversity, and a unique diet-driven microbial signature) in parallel to modulating energy extraction by microbes (as evidenced by higher fecal short chain fatty acids) and host hormonal systems.

Collectively, these observed effects converge to produce on a net result of less energy being available to the human host. We revised the manuscript to be clear that our data is not causal, but that the effects seen in human energy balance were in alignment with the preclinical literature that demonstrates that the gut microbiome contributes to host energy balance by modulating energy available to the host (<https://doi.org/10.1073/pnas.0407076101>; <https://doi.org/10.1038/nature05414>).

We also removed the word “reprogramming” in several locations, including the discussion, to shift the focus in the direction of host-diet-gut microbiome interactions.

- 14) It would be great if you could discuss the energy that is left in faeces.

The energy that is left in feces is defined as the energy in feces normalized to the dietary input: fecal energy loss (g COD/day)/Energy intake (g COD/day). This approximates host metabolizable energy (See Fig. 1b) very closely, as the only missing elements are quantitatively minor contributors (combustible gas, urinary energy, surface energy, and heat released as a result of metabolic inefficiency; <https://doi.org/10.1038/sj.ejcn.1602938>).

The components of fecal energy include undigested food, bacterial biomass, and bacterial metabolites. This review <https://doi.org/10.1080/10643389.2014.1000761> reports: “Bacterial biomass is the major component (25–54% of dry solids) of the organic fraction of the feces. Undigested carbohydrate, fiber, protein, and fat comprise the remainder and the amounts depend on diet and diarrhea prevalence in the population.”

Note: PCR quantification of microbial 16S rRNA gene showed an approximately 8.5-fold increase in bacterial 16S rRNA gene copies per day.

Combined, we conclude that the diet effects on energy balance, which is solely driven by fecal energy loss (not changes in energy expenditure), are modulated in part by the gut microbiome (mass and energy harvest). We have clarified these points in the discussion (lines 287-298) as follows:

Microbial communities in the gut have a profound impact on mammalian host endocrinology, physiology, and energy balance, with most causal inferences historically restricted to preclinical animal models^{5,7,8}. Prior human studies exploring the relationships among the gut microbiome, obesity and energy balance lacked the deep phenotyping, precise methodologies, and rigorous controls that are instrumental for drawing causal inferences with respect to human health. Our central finding was that a diet designed to feed and modulate the colonic gut microbiome, under conditions of fixed energy intake and physical activity, led to reduced metabolizable energy by the host due to increased fecal energy output consisting of undigested food, bacterial biomass, and microbial metabolites but not to changes in energy expenditure. Thus, the greater fecal energy loss on the MBD was not just

due to undigested food, but also to an increase in fermenting gut microbes and their metabolites. Although higher energy harvest by microbes is believed to lead to more energy being absorbed by the host based primarily on preclinical models⁸, our results show the opposite: host metabolizable energy was lower due to higher fecal energy loss on the MBD.

- 15) Are the differences observed between dietary interventions due to differences in energy extraction or differences in bacterial biomass?

There is solid evidence for both. Energy extraction is increased, as evidenced by higher fecal and serum SCFAs (Fig. 2 c and d). In addition, biomass goes up approximately 8.5-fold (see Fig. 2a- note log scale). Importantly, the energy in biomass was modeled to make up >25% of the total fecal COD. We have clarified this as follows:

Results (lines 120-124)

Mean daily fecal weight was higher on the MBD ($P < 0.0001$; Supplementary Fig. 3a), and a proportion of this additional weight was due to a significant increase in 16S rRNA genes ($P < 0.0001$; Fig. 2a), an indication of fecal bacterial biomass increase since the MBD was predicted by our *in silico* mathematical model¹⁵ to produce 19.6 ± 3.5 gCOD/d of microbial biomass compared to 9.4 ± 1.2 gCOD/d on the WD, which is >25% of the total energy content of feces.

Discussion (lines 326-334)

The quantitative contributions of gut microbes to host energy balance were addressed in two ways. First, the microbial biomass was modeled to contribute to >25% of the total fecal energy on the MBD. Second, the fermentation increased as evidenced by increased fecal and serum SCFAs on the MBD as compared to the WD. Thus, host energy absorption shifted towards microbially produced SCFAs and away from proximally digested and absorbed nutrients. While the quantitative contribution of microbially generated SCFAs as inputs to host energy balance was negated by the additional loss of microbial biomass in the feces, the uptake of more microbially produced SCFAs was associated with increased total GLP-1 and pancreatic polypeptide concentrations which may trigger important energy homeostasis signaling cascades to promote satiety and suppress hunger^{47,48}.

- 16) The authors argue that butyrate-producing bacteria responded to the dietary intervention. Did you observe any correlations between the bacterial group responding to the intervention and changes in SCFAs? *The relationship of butyrate concentration and butyrate-producing bacteria is complex because the butyrate concentration in the feces is determined by a complex interaction among butyrate production by bacteria, butyrate consumption by other bacteria, and butyrate absorption by the host. Given the complexity inherent to butyrate dynamics, a simple correlation of butyrate concentration is not meaningful, and this is what we found. We are working on a detailed analysis on microbial interactions which would include production and consumption of SCFA on a separate paper.*

We modulated our statements in the discussion to make it evident that the current paper poses the role of butyrate as a hypothesis (lines 336-343):

We also found a taxonomic signature that was in alignment with the expected impacts of the substrates available to the gut microbes on the two diets. Many of the species detected at higher abundance on the MBD were fiber degraders and/or butyrate producers. We posit that the higher relative abundance of microbes that produce SCFAs could modulate several components of the energy balance equation.

- 17) Line 132-136: Based on the changes in the gut microbiome, how did you conclude that the microbiota signature that defined the response to the MBD channeled more energy to the microbes (instead of the host) whereas the WD led to conditions in which the gut microbes were “starved”??

We eagerly take this opportunity to elaborate on this topic, which is one of the take-home messages of our work. The concept of “starving” the microbiome was introduced by Sonnenburg (<https://doi.org/10.1016/j.cmet.2014.07.003>) and refers to the negative impact of diets lacking in “microbiota accessible carbohydrates” on gut microbiome composition and function. In our experimental design, the Western Diet (WD) was low in microbiota accessible carbohydrates while the Microbiome Enhancer Diet was high. This forms the basis of our ability to contrast a gut microbiome that receives relatively little substrate from the

host diet (“starved”) with one that receives a much higher concentration of dietary substrates. This occurred within each host because of our crossover design and was not confounded by differences dietary energy or macronutrients.

All of this now summarized in the discussion in lines 375-384, which are new:

Our results collectively indicate that when dietary substrates are less available to the gut microbes (as with the WD), the microbes are “starved” of host diet-derived substrates. This is in agreement with the findings from Sonnenberg et al¹⁴. The lower 16S rRNA gene copy number on the WD illustrate a decrease in microbial biomass due in part to lower fermentable substrate availability to the host from the diet. Lower fecal and serum SCFAs on the WD point to lower microbial fermentation, which is an indicator of reduced microbial energy harvest⁸. In addition, the increased relative abundance of mucin degraders on the WD suggests that the microbiota was “starved” of diet-derived substrates and turned to the host-derived energy sources like mucin. Mouse models have shown that *B. thetaiotaomicron*, which normally degrades glycans from plant-based foods, consumes host-derived mucin when diet-derived glycans are not available in sufficient quantities⁵⁸. Similarly, mucin-degrading bacteria are more abundant in humans on calorie restricted diet or suffering from anorexia⁵⁹.

Minor comments

- 1) Please specify throughout the manuscript where the hormones were measured. I assume it was measured in blood. *We apologize for this omission. All the hormones were measured in blood, and this detail was added to the methods and results.*
- 2) It would encourage the authors to upload their sequencing data to public repositories. *This has been done, here are the reviewer links (public links will be provided in manuscript before acceptance):*

Raw Data:

Accession Number: PRJNA913183

Reviewer Link: <https://dataview.ncbi.nlm.nih.gov/object/PRJNA913183?reviewer=4hqcm2cmouvvegs8qpcimrv9ml>

Processed Data:

Accession Number: PRJNA947193

Reviewer Link: <https://dataview.ncbi.nlm.nih.gov/object/PRJNA947193?reviewer=1a7nf5i98qgbu4covrpg66gckh>

- 3) The scale of the colour-gradient showing relative abundance (RA) in figures is a little unclear to me. Maybe you can explain/specify this better (see Fig 2, see Fig 4).

Thanks for pointing this out. We did not mention that the gradient in those figures is based on log-transformed relative abundance. We now include this information in the figures and legends. The Revised figures and legends are below with added text highlighted in yellow. Supplementary figures (not shown) have been edited in the same manner.

Fig 2. Diet reprogrammed the gut microbiome. e, Heatmap showing the natural-log-transformed mean relative abundances of species whose relative abundances were significantly different by diet; bar plot shows the effect size of

the regression coefficient from compound Poisson regression models comparing the relative abundance of each species by diet.

Fig. 4. The contributions of the gut microbiome to host energy harvest. a, Heatmap showing the associations between host ME and the **natural-log-transformed** mean species relative abundance.

Reviewer 3

Thank you for the opportunity to review the paper by Smith et al. This was a highly controlled and rigorous cross-over feeding trial of a Western Diet and Microbiome Enhancer Diet among healthy weight and overweight young to middle-aged adults in the Southeastern U.S. The primary goal was to examine the effect of the diets on the gut microbiome, energy harvest, and host-related factors pertaining to adiposity and enteroendocrine systems. The study demonstrates in an elegant way, host-diet-microbiome interaction. The primary outcome demonstrates that the Microbiome Enhancer Diet is associated with greater fecal energy loss. This finding is relevant to obesity prevention and control. The study contributes significantly to the field given few if any, human dietary intervention trials have been conducted with such rigor to precisely understand diet's effect on the gut microbiome and related host responses.

We appreciate your favorable impression of our work.

- 1) In this Reviewer's opinion, additional information pertaining to the participant's baseline diet pattern and how closely it aligns with the Western or Microbiome Enhancer Diet is relevant.

This is an important question. We respectfully underscore that the free-living diet was not the "baseline" given our experimental design, which measured endpoints after consumption of the study diets. Importantly, the a priori comparison was within-subject differences between the control Western Diet and the Microbiome Enhancer diet.

*To directly answer your question, we evaluated the participant's free-living self-reported dietary intake using the NCI Diet History II questionnaire, which provides self-reported intake data over the previous 12 months. This tool allowed us to evaluate dietary fiber, which is one of the four drivers of our dietary intervention. The 17 participants enrolled in our study reported consuming an average of 7.6g/1000 kcal/d of total fiber (IQR 6.6, 10.4g/1000 kcal/d). Although we recognize that food frequency questionnaires may underestimate intake, these levels of total fiber intake are similar to those reported by ~8,000 adult NHANES participants without diabetes (8.7 ± 0.11 g/1000 kcal/d for females and 7.7 ± 0.11 g/1000 kcal/d for males) (<https://doi.org/10.1017/S0007114523000089>). Therefore, our participants had a low free-living fiber intake, on average, which is on par with the average American and similar to the study-provided Western diet (mean +/- SEM 6.4 ± 0.02 g total fiber/1000 kcal/d), as shown in **Supplementary Table 3**.*

*The free-living participant-reported distributions of calories consumed as carbohydrate, fat, and protein were (mean +/- SEM: 51 +/- 3% carbohydrate, 34 +/- 2% fat, and 16 +/- 2%) were similar to the macronutrient ranges of the controlled study diets (47-52% carbohydrate, 32-37% fat, and 15-18% protein, **Supplementary Table 3**).*

*We added a description of the methods to lines 485-490, and these nutritional parameters to **Supplementary Table 3** and in the results (see lines 90-93)*

Free-living dietary intake. Outpatient dietary intake was evaluated using the validated food frequency questionnaire, the Diet History Questionnaire II (Diet History Questionnaire, Version 2.0. National Institutes of Health, Epidemiology and Genomics Research Program, National Cancer Institute. 2010), on which patients self-reported the frequency and portion sizes of food items consumed over the past 12 months. Diet Calc software was used to estimate nutrients

consumed based upon the United States Department of Agriculture food database (Diet*Calc Analysis Program, Version 1.5. National Cancer Institute, Applied Research Program).

2) Moreover, sleep-related data is also relevant to characterizing the findings.

*Free-living sleep data was analyzed [from the armband accelerometers worn by participants prior to the start of Period A. We now describe this briefly in a newly added methods section titled “Armband accelerometry.” The mean \pm s.e.m. sleep duration was 5.95 ± 0.32 hours. This is now added to **Supplementary Table 4**.*

Sleep time whilst in the whole room calorimeter was measured by movement sensors in the whole room calorimeters over 6 days and was not different by diet. Details were added to the manuscript as follows:

Results (lines 70-79)

Surveillance of adverse events revealed minimal gastrointestinal or other side effects (Supplementary Table 1). Adherence was equivalent between diets during the metabolic ward period ($99.6 \pm 0.19\%$ on MBD vs. $99.9 \pm 0.10\%$ on WD, $p = 0.27$; Supplementary Fig. 2b). The total sleep period during the 6-days in the calorimeter (when our primary endpoint was measured) was held constant between diets (8-hours; Supplementary Table 2), which is important because sleep duration impacts hunger, circadian rhythms and downstream host and microbial phenotypes¹⁶. With our radar motion detector, we calculated motion-free sleep, which is a surrogate of high-quality sleep that we use to minimize the effects of small amounts of involuntary motion during sleep on sleep energy expenditure¹⁷. We found that sleep was not different between the two diet conditions during the calorimetry stay (mean \pm s.e.m. motion-free sleep duration was 3.5 ± 0.75 hours on WD and 3.5 ± 0.5 hours on MBD; Supplementary Fig. 2c).

Supplementary Fig. 2. The experimental paradigm achieved adherence and energy balance. c, Motion-free sleeping time in hours (mean of 6 measurement days) calculated from the whole-room calorimeter radar sensor by removing all minutes with ≥ 6 counts of movement.

Methods (lines 439-446)

Whole-room indirect calorimetry. Energy expenditure and all its subcomponents was evaluated every 24-hours with whole room calorimetry in two 6-day blocks per diet (days 24-29 and 53-58¹²) following published standards of operation⁶⁴. Activity was tightly controlled during the day to maintain spontaneous physical activity consistent within and between participants and to ensure consistent times of meals, exercise, type of activity, rest and sleep¹². Motion-free sleep was calculated from the radar motion detector in the calorimeter by removing all minutes with ≥ 6 counts of movement. This is a surrogate of high-quality sleep that we use to minimize the effects of small amounts of involuntary motion during sleep on sleep energy expenditure¹⁷. Supplementary Table 2 demonstrates the calorimetry schedule.

REVIEWERS' COMMENTS

Reviewer #1 (Remarks to the Author):

The reviewers have addressed my concern regarding the possible interim look and the wash out period. They have also corrected some misconceptions I had. However, I continue to have concern regarding a couple aspects.

First, as noted previously, the sample size is modest. While inclusion of strat factors is important, it remains the case, nonetheless, that modest sample size prevents many of the nice properties underlying randomization from kicking-in appropriately. This is not a deal breaker, but it is still something that really should be better acknowledged.

Second, the use of maaslin should be verified via more classical methods as it (and most biobakery tools) have many hidden aspects that lead to unacceptable false positive rates. In this case, the distributional assumptions are fairly strong, but it would be reassuring for the results to be validated using alternative methods that do not make such strong parametric assumptions. I agree that one could lose power, but all hypothesis testing is concerned with protecting type I error first and power second.

Reviewer #2 (Remarks to the Author):

Thanks for a careful and thorough revision, which has greatly improved the manuscript. I only have two minor comments.

The authors report that biomass was an important variable predicting host metabolizable energy. First, the authors should argue or justify why bacterial copy number can be used as a surrogate of biomass. Secondly, I think it is most fair to write throughout the manuscript that bacterial copy number, a surrogate of biomass, was predictive, since the authors did not measure bacterial biomass per se.

I think the authors should consider to include: “: a controlled feeding study” in the title of the manuscript.

Best wishes,

Henrik M. Roager

Reviewer #3 (Remarks to the Author):

Thank you for the opportunity to review the revised manuscript by Smith et al. "Human Host-Diet-Gut Microbiome Interactions Influence Energy Balance." The authors addressed my specific concerns (free-living dietary data and sleep-related data) and were thoughtful in their responses to Reviewers 1 and 2. The authors need to check their rounding across tables given rounding errors - for example Suppl Table 4 - % kcal macronutrients. Also, give some context to what Bristol Type 3 and 4 refer to in the footnotes. I think with minor editorial revisions, the manuscript is suitable for publication.

May 8th, 2023

Manuscript Number: NCOMMS-23-02679B

Point by Point Response to Reviewer Comments:

We are thankful for the helpful second round of feedback from the reviewers and are pleased that our initial revisions greatly improved the manuscript.

Our replies to reviewers are presented in blue font, below. In our answers, we indicate the exact text that was changed in the manuscript. Revisions to the manuscript are marked with yellow highlighting. Throughout our responses, below, we provide the surrounding text for context when appropriate.

The revised manuscript satisfies all editorial and reviewer requirements, and we are pleased to submit this final document for publication.

Reviewer 1

The reviewers have addressed my concern regarding the possible interim look and the wash out period. They have also corrected some misconceptions I had. However, I continue to have concern regarding a couple aspects.

We are pleased that we were able to address many of your important recommendations.

- 1) First, as noted previously, the sample size is modest. While inclusion of strat factors is important, it remains the case, nonetheless, that modest sample size prevents many of the nice properties underlying randomization from kicking-in appropriately. This is not a deal breaker, but it is still something that really should be better acknowledged.
Although randomization for balancing of covariates and equivalent treatment allocation is indeed challenging in some designs, including those with small sample sizes¹, this is not a great concern in our design because each participant serves as his/her own control, and we stratified by sex. Wellek et al. state that “The essential feature distinguishing a crossover trial from a conventional parallel-group trial is that each proband or patient serves as his/her own control. The crossover design thus avoids problems of comparability of study and control groups with regard to confounding variables (e.g., age and sex). Moreover, the crossover design is advantageous regarding the power of the statistical test carried out to confirm the existence of a treatment effect: Crossover trials require lower sample sizes than parallel-group trials to meet the same criteria in terms of type I and type II error risks.²

We are confident that randomization was appropriate and have added the following details to the manuscript to clarify the approach:

(Lines 451-459)

Design overview. This was a randomized crossover study with a control Western Diet (WD) compared to a Microbiome Enhancer Diet (MBD) where each participant served as their own control, **thereby minimizing the impact of confounders². We applied block randomization stratified by sex. The randomization code was generated by the study statistician who worked directly with the study dietitian in charge of assigning menus to participants. Participants were enrolled by the study coordinator. In order to balance sex, we randomly assigned 3 blocks to each sex. Within each block (n=6), participants were randomly assigned using simple randomization to one of two diet sequences with a 1:1 allocation ratio using SAS PROC PLAN. Eight participants were randomized to sequence 1 (WD followed by MBD) and 9 participants were randomized to sequence 2 (MBD followed by WD).**

- 2) Second, the use of maaslin should be verified via more classical methods as it (and most biobakery tools) have many hidden aspects that lead to unacceptable false positive rates. In this case, the distributional assumptions are fairly strong,

but it would be reassuring for the results to be validated using alternative methods that do not make such strong parametric assumptions. I agree that one could lose power, but all hypothesis testing is concerned with protecting type I error first and power second.

Thank you for your suggestion. We validated our results related to differential abundance by diet with ANCOM-BC, and approach often used to evaluate differential abundance and has been found to produce results that frequently overlap with other approaches³. It does not make strong distributional assumptions^{4,5} and is reported to adequately control the false discovery rate as compared to other methods^{3,5}. It also handles continuous fixed variables and random variables, which means that we can be consistent with our linear mixed model parameters⁵. A brief summary of this approach is now in the methods section.

ANCOM-BC determined that five species were differentially abundant by diet: *Oscillibacter* sp. 5720, *Eubacterium eligens*, *Roseburia hominis*, *Lachospira pectinoschiza*, and *Prevotella copri*. All the species, apart from *R. hominis*, were among the significantly differentially abundant species detected by MaAsLin2 with q-values < 0.05 and effect sizes ≥ 2 . *R. hominis* had a q-value of 0.0002, but an effect size under 2. ANCOM-BC did not determine *Oscillibacter* sp. CAG 241 or *Prevotella* sp. CAG 279 to be differentially abundant, while MaAsLin2 did. Overall, 8 of the 10 species in our MaAsLin2 analysis were near the top of the ranking by ANCOM-BC, with 7 of 10 having significant P-values and 4 of the 10 having significant Q values. Therefore, we are confident in the MaAsLin2 results.

To strengthen our manuscript by demonstrating reproducibility, we have revised the text as follows:

(Lines 164-170)

In contrast, the 4 species with a higher relative abundance on the WD included *Blautia hydrogenotrophica*, *Bifidobacterium pseudocatenulatum*, uncharacterized *Blautia* CAG:257, and uncharacterized *Actinomyces* ICM7 (Q = 0.006, 5.6×10^{-05} , 0.001, 0.02, respectively). These four species derive their source of fermentation from host-glycans, simple sugars^{6,7}, or fermentation products generated by other gut microbes, mainly CO₂⁸ and H₂⁹. As a means of validation, we repeated this analysis with ANCOM-BC^{3,5}, and retained the significance of most of the identified species in the signature (Supplementary Table 5).

For the validating correlations between relative abundance and metabolizable energy on the microbiome enhancer diet (MBD) only, we employed Kendall's tau-b correlation coefficient¹⁰ because it is a non-parametric test that is often used to explore the relationship between relative abundance and continuous or ordinal variables¹¹. We used the same parameters as we did with Maaslin2: 1) prevalence filtering at 25%, and 2) correction for multiple comparisons with Benjamini-Hochberg method. A brief summary of this approach is now in the methods section.

After analyzing differential abundance by host metabolizable energy in only MBD samples and correcting for multiple comparisons via Benjamini-Hochberg and filtering species for 25% prevalence, Kendall's correlation test did not find any significantly associated species. This within-diet comparison lacked the power afforded by the crossover design, and we believe that this is the primary reason for the observed results. Because we were unable to validate these results, we present significantly abbreviated results as hypothesis-generating and requiring validation in future studies designed and powered to explore the factors that contribute to the variability in metabolizable energy when there is increased delivery of dietary substrates to the gut microbiota.

To avoid overinterpretation of our hypothesis-generating findings, we removed these data from the abstract and discussion, removed the figure from the main text, and toned down our presentation of the results as follows:

(Lines 261-272)

This led us to postulate that the quantitatively important variability in host energy balance could be associated with the repertoire of gut microbes in the colon. To test this, we asked whether the variability in host metabolizable energy on the MBD could be related to a unique microbial signature. To identify those microbial signatures, we derived regression coefficients describing each microbe's association with the independent variable of host metabolizable energy using MaAsLin2's compound Poisson regression model¹². In total, host metabolizable energy was associated with the relative abundance of 16 species (Supplementary Fig. 7a-b). Four of those species had Q <

0.05 and effect size ≥ 2 and have been identified as differentially abundant after weight loss (due to bariatric surgery¹³ or caloric restriction¹⁴) and in bile acid metabolism¹⁵, suggesting a potential role in weight regulation. Our results were not reproducible with an alternative method (Kendall's tau-b correlation¹⁰; Supplementary Table 6) and should be considered hypothesis-generating. Future studies that are designed and powered to explore the microbial species that explain the variability in host ME on the MBD are needed to confirm these results.

Reviewer 2

Thanks for a careful and thorough revision, which has greatly improved the manuscript. I only have two minor comments.

- 1) The authors report that biomass was an important variable predicting host metabolizable energy. First, the authors should argue or justify why bacterial copy number can be used as a surrogate of biomass. Secondly, I think it is most fair to write throughout the manuscript that bacterial copy number, a surrogate of biomass, was predictive, since the authors did not measure bacterial biomass per se.

We appropriately defined that 16S rRNA copy number is a surrogate of biomass (and not biomass per se) in the abstract, upon first use in the results, and upon first use in the discussion.

- 2) I think the authors should consider to include: “: a controlled feeding study” in the title of the manuscript. Great idea. We included “a Randomized Clinical Trial” per editorial request, to meet the required 15-word limit, and to align with CONSORT guidelines. The revised title is:

Host-Diet-Gut Microbiome Interactions Influence Human Energy Balance: a Randomized Clinical Trial

Reviewer 3

Thank you for the opportunity to review the revised manuscript by Smith et al. "Human Host-Diet-Gut Microbiome Interactions Influence Energy Balance." The authors addressed my specific concerns (free-living dietary data and sleep-related data) and were thoughtful in their responses to Reviewers 1 and 2.

- 1) The authors need to check their rounding across tables given rounding errors - for example Suppl Table 4 - % kcal macronutrients.

Thank you for catching this. We have reviewed all rounding and corrected errors. Note that in the table you specifically call out, the % of kcals from macronutrients does not add up perfectly to 100% because the kcals are calculated from energy values using constants that represent an average of energy per gram of carb, protein or fat.

- 2) Also, give some context to what Bristol Type 3 and 4 refer to in the footnotes.

We added the following sentence to Supplementary Table 4 (which we now included in the main paper as Table 1):

The Bristol Stool Scale evaluates stool type based on shape and consistency, with scores of 3-4 indicating neither constipation nor loose stools.

I think with minor editorial revisions, the manuscript is suitable for publication.

Thank you!

References

- 1 Suresh, K. An overview of randomization techniques: An unbiased assessment of outcome in clinical research. *Journal of human reproductive sciences* **4**, 8-11, doi:10.4103/0974-1208.82352 (2011).
- 2 Wellek, S. & Blettner, M. On the proper use of the crossover design in clinical trials: part 18 of a series on evaluation of scientific publications. *Deutsches Arzteblatt international* **109**, 276-281, doi:10.3238/arztebl.2012.0276 (2012).
- 3 Nearing, J. T. et al. Microbiome differential abundance methods produce different results across 38 datasets. *Nature Communications* **13**, 342, doi:10.1038/s41467-022-28034-z (2022).
- 4 Lin, H. & Peddada, S. D. Analysis of compositions of microbiomes with bias correction. *Nature Communications* **11**, 3514, doi:10.1038/s41467-020-17041-7 (2020).
- 5 Lin, H. & Peddada, S. D. Analysis of microbial compositions: a review of normalization and differential abundance analysis. *npj Biofilms and Microbiomes* **6**, 60, doi:10.1038/s41522-020-00160-w (2020).
- 6 Cordeiro, R. L. et al. N-glycan utilization by bifidobacterium gut symbionts involves a specialist β -mannosidase. *Journal of Molecular Biology* **431**, 732-747, doi:<https://doi.org/10.1016/j.jmb.2018.12.017> (2019).

- 7 Wu, C. in *Encyclopedia of Metagenomics* (ed Karen E. Nelson) 1-7 (Springer New York, 2013).
- 8 Liu, X. et al. Blautia—a new functional genus with potential probiotic properties? *Gut microbes* **13**, 1875796, doi:10.1080/19490976.2021.1875796 (2021).
- 9 Smith, N. W., Shorten, P. R., Altermann, E., Roy, N. C. & McNabb, W. C. Examination of hydrogen cross-feeders using a colonic microbiota model. *BMC Bioinformatics* **22**, 3, doi:10.1186/s12859-020-03923-6 (2021).
- 10 Schaeffer, M. S. & Levitt, E. E. Concerning Kendall's tau, a nonparametric correlation coefficient. *Psychological bulletin* **53**, 338-346, doi:10.1037/h0045013 (1956).
- 11 Fouladi, F. et al. Air pollution exposure is associated with the gut microbiome as revealed by shotgun metagenomic sequencing. *Environment International* **138**, 105604, doi:<https://doi.org/10.1016/j.envint.2020.105604> (2020).
- 12 Mallick, H. et al. Multivariable association discovery in population-scale meta-omics studies. *PLOS Computational Biology* **17**, e1009442, doi:10.1371/journal.pcbi.1009442 (2021).
- 13 Palmisano, S. et al. Changes in gut microbiota composition after bariatric surgery: a new balance to decode. *J Gastrointest Surg* **24**, 1736-1746, doi:10.1007/s11605-019-04321-x (2020).
- 14 Snierski-Kind, J. et al. Effects of caloric restriction on the gut microbiome are linked with immune senescence. *Microbiome* **10**, 57, doi:10.1186/s40168-022-01249-4 (2022).
- 15 Song, Y. et al. *Clostridium bolteae* sp. nov., isolated from human sources. *Systematic and Applied Microbiology* **26**, 84-89, doi:<https://doi.org/10.1078/072320203322337353> (2003).